# Deep microbial colonization during impact-generated hydrothermal circulation at the Lappajärvi impact structure, Finland

Jacob Gustafsson [1] ✉, Gordon R. Osinski [2], Nick M. W. Roberts [3], Jay Quade[4], Zhennan Wang[4], Martin J. Whitehouse [5], Heejin Jeon[5], Andreas Karlsson [5], Satu Hietala [6] & Henrik Drake [1] ✉

Deeply fractured rocks of meteorite impact structures have been hypothesized as hot spots for microbial colonization on Earth and other planetary bodies. Biosignatures of such colonization are rare, however, and most importantly, direct geochronological evidence linking the colonization to the impact-generated hydrothermal systems are completely lacking. Here we provide timing constraints to microbial colonization of the $77.85 \pm 0.78$ Ma old Lappajärvi impact structure, Finland, by using coupled microscale stable isotope biosignature detection and radioisotopic dating of vug- and fracture-filling assemblages in impactites. The first detected mineral precipitation at habitable temperatures for life ($47.0 \pm 7.1\,°C$) occurred at $73.6 \pm 2.2$ Ma and featured substantially $^{34}S$-depleted pyrite consistent with microbial sulfate reduction. Later stages of vug-mineral precipitation occurred more than 10 Myr later, at gradually lower temperatures, and featured $\delta^{13}C_{calcite}$ values diagnostic for both anaerobic microbial consumption and production of methane. These insights confirm the capacity of medium-sized (and large) meteorite impacts to generate long-lasting hydrothermal systems, enabling microbial colonization as the crater cools to ambient conditions, an effect that may have important implications for the emergence of life on Earth and beyond.

Meteorite impact craters occur on all planetary bodies in our Solar System. Once thought to be purely destructive events, there is a growing consensus that meteorite impacts may have played an important role in the origin and early evolution of life on Earth[1,2]. An impact event deposits a tremendous amount of energy in a planetary crust, which results in the fracturing and heating of a huge volume of rock that scales with crater size. Fractured rocks within impact structures are considered key candidates for deep microbial colonization on planetary bodies due to their sustained heat flow, the formation of

impact-generated hydrothermal (IGH) systems with distinct geochemical and thermal gradients, and the presence of pore spaces that support microbial colonization[1,2]. However, biosignatures of microbial colonization have been reported from only a handful of the more than 200 confirmed impact structures on Earth[3,4].

Impact events create hydrothermal systems that initially exceed temperatures that hyperthermophiles can survive (up to $122\,°C$[5]), requiring cooling prior to any microbial colonization[6,7]. The time needed to reach habitable conditions depends on the size of the

---

[1]Department of Biology and Environmental Science, Linnaeus university, Kalmar, Sweden. [2]Department of Earth Sciences, University of Western Ontario, London, ON, Canada. [3]Geochronology and Tracers Facility, British Geological Survey, Nottingham, UK. [4]Department of Geosciences University of Arizona, AZ Tucson, USA. [5]Department of Geosciences, Swedish Museum of Natural History, Stockholm, Sweden. [6]Geological Survey of Finland, Kuopio, Finland. ✉e-mail: jacob.gustafsson@lnu.se; henrik.drake@lnu.se

impact structure. For example, modeling suggests that the 4 km diameter Kärdla impact structure could become habitable for microbial communities within tens to a few hundreds of years post-impact[8]. Larger impact structures, which undergo significant melt production and deep heat sterilization, take longer to become habitable. The scale of the impact, along with other factors[2] determines the longevity of IGH systems. Large craters like the ~250 km Vredefort and Sudbury structures as well as in the ~200 km Chicxulub structure, may have hosted persistent IGH systems for hundreds of thousands of years, up to 1–2 million years[9–11]. In the latter structures, near-surface regions feature effective convective heat transport, while deeper levels are marked by dominantly conductive heat transport[6]. Midsized to large structures can host prolonged IGH systems if they are added heat to near-surface parts by meteoric water and groundwaters that are heated at back-circulated from several kilometers depth after infiltration along faults and fractures[6,9]. High-precision U-Pb dating[12] and $^{40}Ar/^{39}Ar$-based thermal modeling[13] of the 23 km diameter-sized and 77.85 ± 0.78 Ma old, Lappajärvi impact structure in Finland, suggests it has experienced hydrothermal circulation and post-impact cooling below 250 °C as late as ~1 to 1.6 Myr after the impact. Duration of this magnitude is more extensive than thermal modeling predictions for two other impact structures: (1) the Ries impact structure (24 km diameter), where the IGH system ceased in less than 250,000 years[14], and (2) the Haughton impact structure (23 km diameter), where temperatures enabling thermophiles and hyperthermophiles to thrive (50–120 °C) lasted over 50,000 years[15]. The extended duration of the hydrothermal system at Lappajärvi would have provided ample time for microorganisms to colonize, making it a prime target for exploring biosignatures of ancient microbial life. However, no evidence of microbial colonization at Lappajärvi has previously been reported.

Putative body and/or trace fossils of microbes, some more ambiguous than others, have been reported in impact materials from a handful of impact structures: Ries[16,17], Chesapeake Bay[18], Siljan[19] and Dellen[20]. Stable isotopic biosignatures indicating post-impact microbial activity have been identified in several terrestrial impact structures. At Haughton, pyrite and marcasite in impact breccias with $\delta^{34}S$ values as low as $-42 \pm 2‰_{V-CDT}$, suggest microbial sulfate reduction (MSR)[21]. Likewise, at Rochechouart, $\delta^{34}S$ sulfide values down to $-36‰$ are consistent with MSR[22]. In the Siljan structure, fracture-coating calcite with $\delta^{13}C$ values ranging from $-52$ to $22‰_{V-PDB}$ is indicative of microbial methanotrophy and methanogenesis[23], potentially in syntrophy with fungi[19]. Pyrite $\delta^{34}S$ values in the same fractures were as low as $-42‰$, suggesting MSR[23]. At the Chicxulub impact crater, microscopic pyrite framboids with $\delta^{34}S$ values as low as $-35‰$ and $\Delta S_{sulfate-pyrite}$ values up to $54‰$ are also consistent with MSR[24], and studies of extant microbiological communities show that the impact-induced geochemical boundaries have shaped the modern-day deep biosphere in the granitic basement underlying this crater[25].

Although these biosignatures have been reported from a limited number of impact structures, they provide valuable evidence of microbial activity following impact events. However, the timing of microbial colonization with respect to these impact events remains uncertain. In other words, there is a possibility that the previously reported biosignatures from impact structures are unrelated to the impact crater formation, and instead represent outcomes of long-term recolonization of the site.

Only a few studies have directly dated the materials hosting or associated with biosignatures. Using in situ Rb/Sr dating of calcite-albite-K feldspar assemblages occurring alongside fossilized fungi, that were initially attributed to colonization during impact-related hydrothermal activity[26], Tillberg et al.[27]. found that fungal colonization of the 458 Ma Lockne impact structure occurred at least 100 million years after the impact. Similarly, micro-scale U-Pb dating of $^{13}C$-enriched methanogenesis-related calcite in fractures and breccias at Siljan yielded ages between 80 and 22 Ma, indicating colonization

occurred 300 million years after the 380.9 ± 4.6 Ma impact event[23]. In the same structure, U-Pb carbonate geochronology provided a maximum age of 39.2 ± 1.4 Ma for coeval fossilized fungi[19]. These findings suggest that documenting biosignatures through petrographic relationships to impact rocks alone cannot confirm impact-related microbial activity. Radioisotopic dating, along with other contexts, such as petrographic relations, is essential to verify such colonization.

In this study, we investigated the Lappajärvi impact structure in Finland to detect ancient biosignatures of microbial colonization in the crater-fill impactites, and to establish timing and temperature constraints. Using micro-scale isotopic techniques, we provide direct evidence of microbial colonization during the waning stages of the IGH system of this terrestrial impact structure. This research has significant implications for the hypothesis that impact craters serve as hot spots for microbial colonization on Earth and, by analogy, on other planetary bodies. This is particularly relevant given that deep biosphere habitats in impact structures are considered favorable targets for Mars exploration for signs of extinct life[28].

## Results

### Petrography and mineralogy in fractures and vugs

Impact melt rock (locally referred to as 'kärnäite'), impact melt-bearing breccia, and lithic (i.e., melt-free) impact breccia, were sampled for vug- and fracture-filling calcite and pyrite, in two cored boreholes (impact melt rock-rich LA1 and impact melt-bearing breccia-rich LA4, Fig. 1). The impact melt rock contains abundant mineral-filled cm-sized vugs at 3–10 m, sporadically around 30 m, and at 144–145 m depth (just above the contact to impact melt-bearing breccia) as well as fractures at 110–140 m depth. The impact melt rock is underlain and surrounded by impact melt-bearing breccia and lithic impact breccia that gradually proceeds downward and outward from the central part of the crater into autochthonous impact breccia, and finally to the fractured bedrock[29]. Several intervals of the impact melt-bearing breccia contained a high proportion of mm- to cm-sized mineral-filled vugs within impact glass of various degrees of alteration (Supplementary Fig. 1, details in Methods). For simplicity, the impact melt rock in LA1 is grouped into upper (3–31 m) and lower (137–145 m) intervals; the impact melt-bearing breccia in LA4 is similarly divided into upper (23–82 m) and lower (142–153 m) intervals.

The vugs and fractures of the impactites have mm to cm thick coatings of secondary minerals (Supplementary Figs. 1, 2) identified via SEM-EDS (Supplementary Figs. 3–6) and X-Ray Diffraction (XRD, Supplementary Fig. 7). including zeolites (chabazite, erionite, and heulandite), siderite (Supplementary Fig. 7a) clay minerals (a poorly crystalline phase with a best XRD-fit to vermiculite, hereafter termed vermiculite* as no certain confirmation could be performed [Supplementary Fig. 7b]), quartz, and Fe(-Ti)-oxides, in addition to calcite and pyrite that are the focus of this study. Veins in the lower impact melt rock feature several generations of mineral precipitation. First, subhedral pyrite crystal aggregates intergrown with calcite (with ~3% Fe, 0.6% Mn, 0.4% Mg) occur in coatings lining the fracture walls (calcite group i, Fig. 2a and Supplementary Fig. 3, and EDS-maps in Supplementary Fig. 6). The calcite is overgrown by vermiculite* and siderite. A younger calcite filling with less Mn content than calcite-i occupies the central, most voluminous part of the vein (calcite ii, Supplementary Fig. 3). In impact melt rock vugs, subhedral Fe-poor calcite is frequent (calcite iii, Fig. 2b and Supplementary Fig. 4), occasionally intergrown with aggregates of subhedral pyrite (Fig. 2a), and occasionally with euhedral pyrite on the calcite crystal surfaces (Supplementary Fig. 2a). Calcite is frequent in vugs of the impact melt-bearing breccia, with scalenohedral crystals (calcite y) with overgrowths (calcite yy) being common in the lower interval, in particular (Fig. 2c and Supplementary Fig. 5, showing slightly higher Mn and Fe concentrations in the younger BSE-bright zone). Zeolites, such as erionite and chabazite (Supplementary Fig. 1a, c), as well as pyrite (Fig. 2d) and Fe-Ni-sulfide

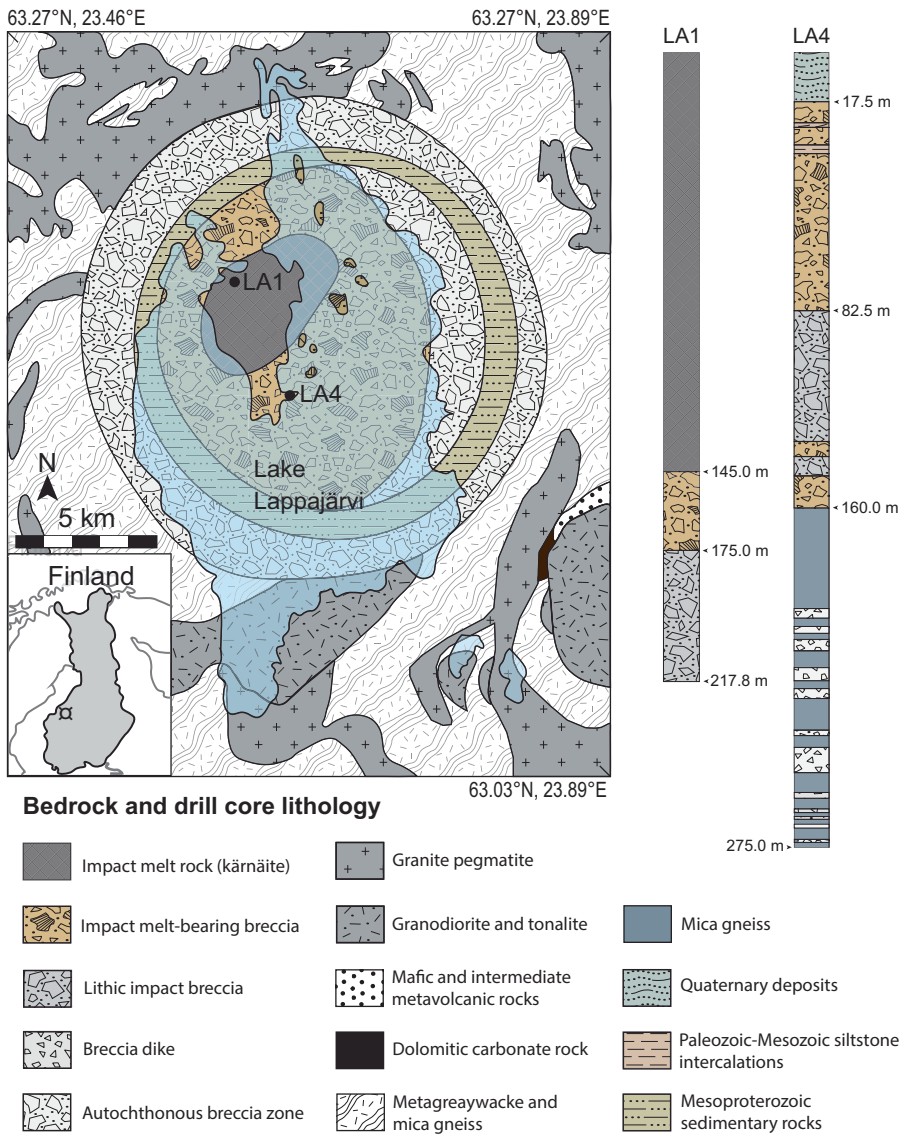

**Fig. 1 | Geological map of the Lappajärvi impact crater, Finland.** Map redrawn from[12] and bedrock and drill core lithology modified after[29,38]. Drill hole locations are marked with LA1 and LA4. Lakes are shown in blue. The coordinates of the map are in decimal degree format using the World Geodetic System 1984 (WGS 84).

aggregates that may contain remnants of the impacting projectile according to previous studies[30], are intergrown with impact glass in the impact melt-bearing breccia. There are also mm-sized euhedral pyrite crystals (Supplementary Fig. 2b) frequently occurring in vugs of the impact melt-bearing breccia and the lithic impact breccia. Occasionally, the latter pyrite occurs adjacent to mm-sized scalenohedral calcite crystals in the lower impact melt-bearing breccia (Supplementary Fig. 2c).

**Stable isotope compositions of pyrite and calcite**
The $\delta^{34}S_{pyrite}$ values of all pyrite crystals analyzed by SIMS in this study range substantially, from −31.9 to 102.2‰$_{V-CDT}$ (Supplementary Data 1). The $\delta^{34}S_{pyrite}$ variability is largest in impact melt rock (−31.9 – 102.2‰). The most $^{34}S$-depleted pyrite (Fig. 3a and Supplementary Fig. 8a) occurs intergrown with calcite group i in the oldest parts of veins of the lower impact melt rock (Fig. 2a). In this sample, there is a trend of increasing $\delta^{34}S_{pyrite}$ values from the crystal cores to rims, by up to 25‰ in single crystals (Fig. 3a and Supplementary Fig. 8a). The most significant small scale $\delta^{34}S$-variability, ranging from −4.3 to 102.2‰ occurred in a polycrystalline pyrite aggregate in a chalcedony-rich interval of the upper impact melt rock (Fig. 3b). Most $\delta^{34}S_{pyrite}$ values in

the impact melt-bearing breccia and lithic impact breccia show small variation, such as in the earliest coarse-grained pyrite with an average of $3.1 \pm 2.1$‰ (1 std. dev.). Two samples with high values, at depths of 29 m (83.2‰, Fig. 3c) and 54 m (76.8‰), make the range substantial. Highly $^{34}S$-enriched pyrite also exists in a lithic impact breccia sample at 130 m, showing increasing values from core to rim ($\delta^{34}S$ of 60.4‰, Supplementary Fig. 8b).

The $\delta^{13}C_{calcite}$ values ranged from significantly $^{13}C$-depleted, −50.9‰, to significantly $^{13}C$-enriched, 16.4‰$_{V-PDB}$, and the $\delta^{18}O_{calcite}$ values from −19.8 to −2.7‰$_{V-PDB}$ (Fig. 4a and Supplementary Data 2). There is a distinct difference in $\delta^{13}C_{calcite}$ values between the impact melt rock (1.7 to 16.4‰) and impact melt-bearing breccia (−52.3 to −14.7‰). Impact melt rock-hosted calcite exhibits homogeneous isotope composition within single crystals (Fig. 4b–d) but clusters into three distinct groups based on their $\delta^{13}C/\delta^{18}O$ composition (Fig. 4a). The group with the highest $\delta^{18}O$ values (group i: −6 to −3‰) is petrographically older (Figs. 2a and 4b, c) than the group (ii) with $\delta^{18}O$ values of −13 to −8‰. The most $^{13}C$-enriched calcite (up to 16.4‰, Fig. 4d) in the impact melt rock is from a coating in a 4 cm diameter vug at 30 m in LA1 (Fig. 2b). The calcite in impact melt-bearing breccia can be divided into two groups, one (group yy) with $\delta^{13}C$ values of −50.9 to −39.8‰

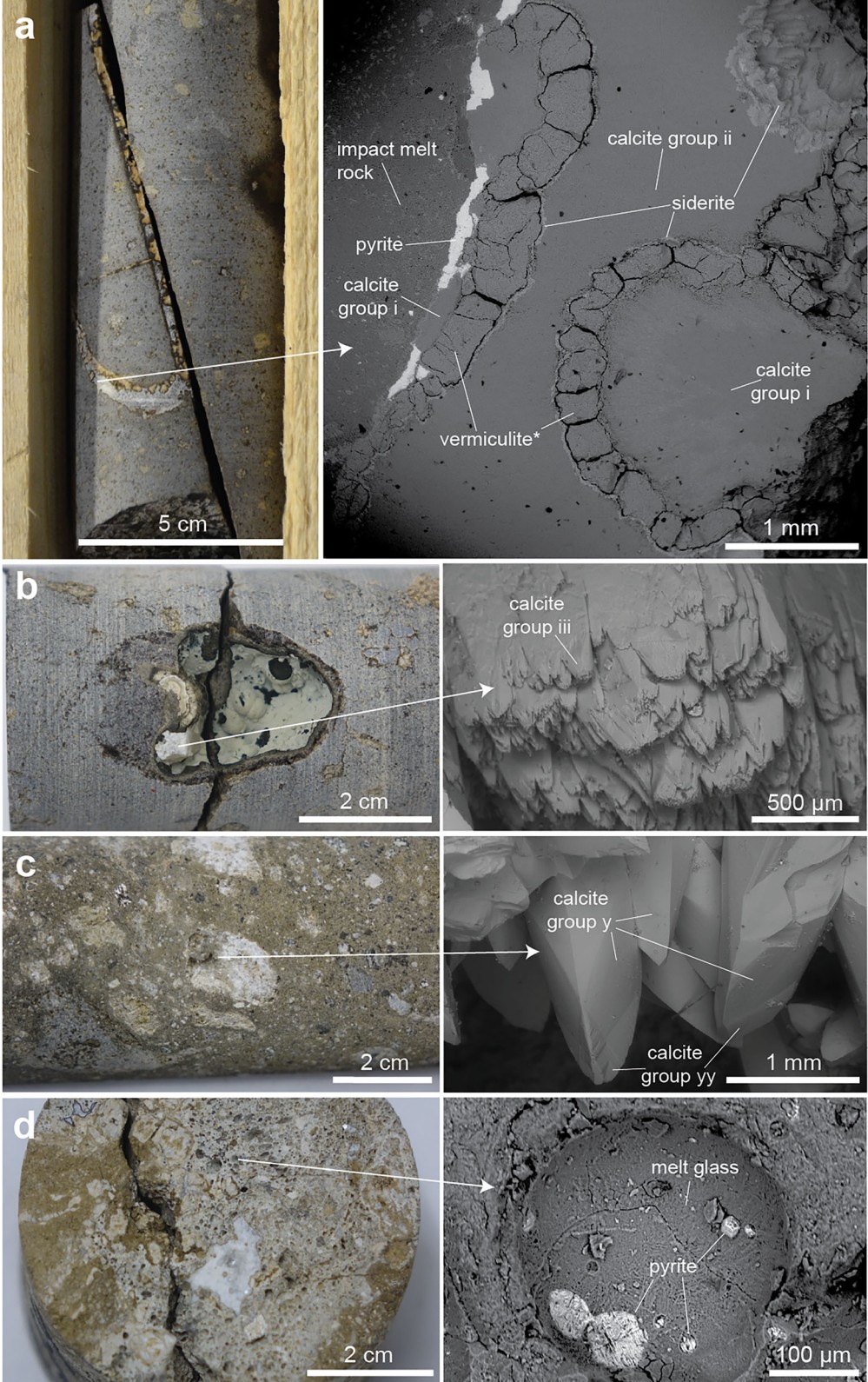

**Fig. 2 | Calcite and pyrite crystals in cavities within impact melt rock and impact melt-bearing breccia.** Drill core photos (left) and mineral details in back-scattered SEM images (right). **a** Sample LA1;137 (drill core ID; drill core length in m): Two generations of calcite (group i and group ii) filling a fracture within impact melt rock. Calcite of group i, adjacent to impact melt rock, is intergrown with pyrite and coated by a poorly crystalline vermiculite-like phase (vermiculite*) and siderite. **b** Sample LA1;30: Subhedral calcite (group iii) in a vug within impact melt rock. **c** Sample LA4;152: Scalenohedral calcite crystals (group y) coating a vug within impact melt-bearing breccia. Some of these crystals have overgrowths of calcite group yy. **d** Sample LA4;59: Pyrite-filled, 10–100 μm circular pits on the wall of a circular vesicles in melt glass of the impact melt-bearing breccia.

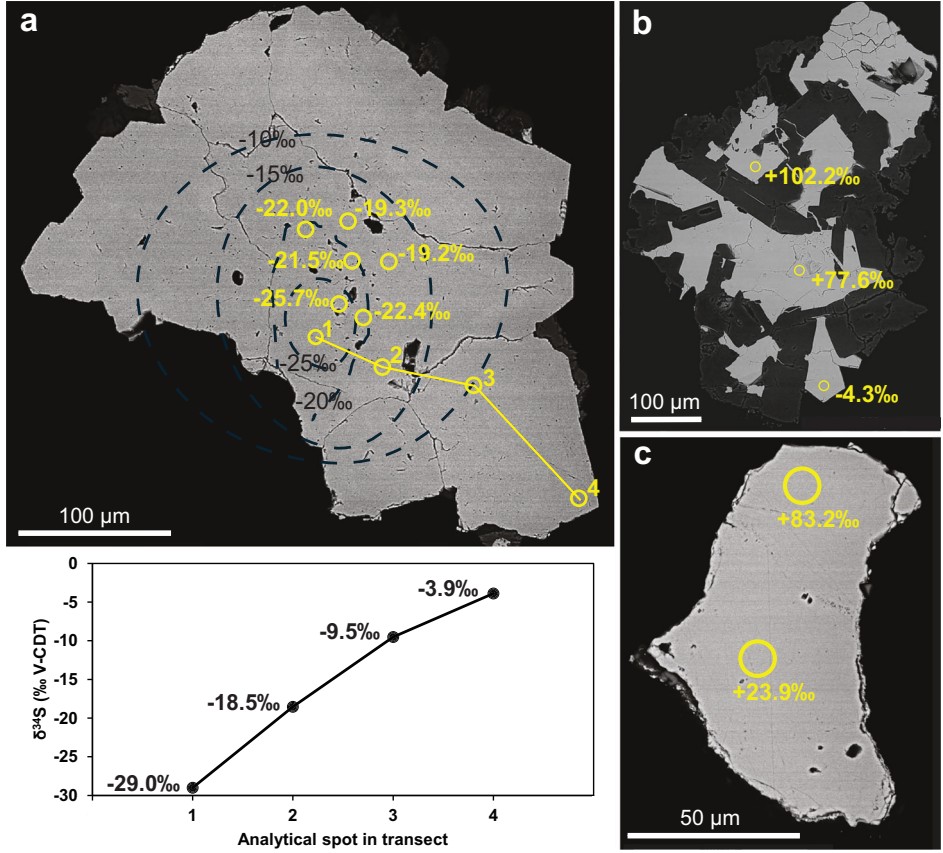

**Fig. 3 | Stable isotope inventory of pyrite.** The SEM images are of polished pyrite crystals in cross-section with corresponding $\delta^{34}S_{pyrite}$ spots based on SIMS analyses. **a** S-isotope transect from core to rim of an $S^{34}$-depleted pyrite crystal in a vein from the impact melt rock sample LA1;137. The diagram for the latter transect shows increased $\delta^{34}S$ values with growth, and interpreted isolines are plotted on the grain for visualization of the spatial distribution of the $\delta^{34}S$ values. **b** $^{34}S$-enriched pyrite in a polymictic grain from the impact melt rock matrix sample LA1;14, showing pyrite along with calcedony (BSE-dark mineral) **c** $^{34}S$-enriched pyrite from a vug in the impact melt-bearing breccia sample LA4;28. The analytical errors (1σ) of the $\delta^{34}S$ values are within the size of the symbols. Each value represents one measurement.

and one with higher values (y: −31.6 to −14.7‰) and with a larger span in $\delta^{18}O$. A large proportion of the impact melt-bearing breccia-hosted calcite exhibits $\delta^{18}O_{calcite}$ values of around −11 to −8‰. The most $^{13}C$-depleted calcite group (yy) only occurs in the lower impact melt-bearing breccia of LA4, and only in the youngest overgrowths of sca-lenohedral calcite crystals in the vugs (Figs. 2c, 4e). The single lithic impact breccia-hosted calcite sample had stable isotope values between the impact melt rock and impact melt-bearing breccia groups (Fig. 4a).

### Clumped isotope compositions of calcite
The clumped isotope analyses yielded Δ638 values which were used for temperature reconstructions (hereafter 'temperatures'), the latter ranging from 8.0 ± 6.8 °C to 47.0 ± 7.1 °C (Supplementary Data 3, 4). Distinct patterns of temperatures are observed between calcite in different $\delta^{13}C/\delta^{18}O$ groups (Fig. 5). The first group of calcite (i) in LA1;137 (cf. Figs. 2a, 4b), has the highest temperature at 47.0 ± 7.1 °C and two other impact melt rock-hosted calcite group i samples show similar, but slightly lower temperatures (Fig. 5a). Two samples from impact melt rock-hosted calcite group ii and iii have temperatures around 30 °C. The calcite samples from the impact melt-bearing breccia, LA4;40 and LA4;152, have temperatures of 8.0 ± 6.8 and 8.9 ± 5.3 °C. The $\delta^{18}O$ composition of the fluids (Supplementary Data 5), from which the calcite precipitated, were calculated based on clumped isotope temperature reconstructions and bulk $\delta^{18}O_{calcite}$ values by applying temperature-dependent fractionation factors[31], showing three distinct $\delta^{18}O_{fluid}$ groups with gradually lower

temperature (Fig. 5). The highest $\delta^{18}O_{fluid}$ values occur in impact melt rock-hosted calcite group i (−2.6 ± 1.6 to −1.0 ± 1.4‰$_{VSMOW}$), followed by impact melt rock-hosted calcite group ii/iii (−8.6 ± 1.1 to −8.1 ± 2.5‰$_{VSMOW}$) and by the impact melt-bearing breccia-hosted calcite samples (−14.2 ± 1.9 to −12.7 ± 1.9‰$_{VSMOW}$).

### U-Pb carbonate geochronology
The two calcite groups i and ii of sample LA1;137 (Figs. 2a and 4b, c) yielded robust ages when targeted for U-Pb geochronology via LA-ICP-MS in the same crystals as previous SIMS-analyses (Supplementary Figs. 3–5). Group i yielded an age of 73.6 ± 2.2 Ma (Fig. 6 and Supplementary Fig. 9) and ii 62.5 ± 3.0 Ma (Fig. 6 and Supplementary Fig. 9). These calcite groups show no discernable individual growth zonation, and their ages are interpreted to represent two single separate pre-cipitation events. Impact melt rock-hosted calcite of the second gen-eration, but with methanogenesis $\delta^{13}C$ signature (group iii, LA1;30, Figs. 2c and 4d), yielded an age that is too imprecise for interpretation (74 ± 28 Ma, Supplementary Figs. 10, 11). The crystal cores (group y) of impact melt-bearing breccia-hosted calcite with AOM signatures in the overgrowths (Figs. 2d, 4d) yielded an age of 43.0 ± 9.6 Ma (Supple-mentary Figs. 9, 11), giving a rough maximum age of the AOM-related calcite.

## Discussion
The highly variable $\delta^{13}C_{calcite}$ and $\delta^{34}S_{pyrite}$ values detected in the vugs and fractures at Lappajärvi show that the geochemical processes, fluid sources, and microbial activities leading to calcite and pyrite

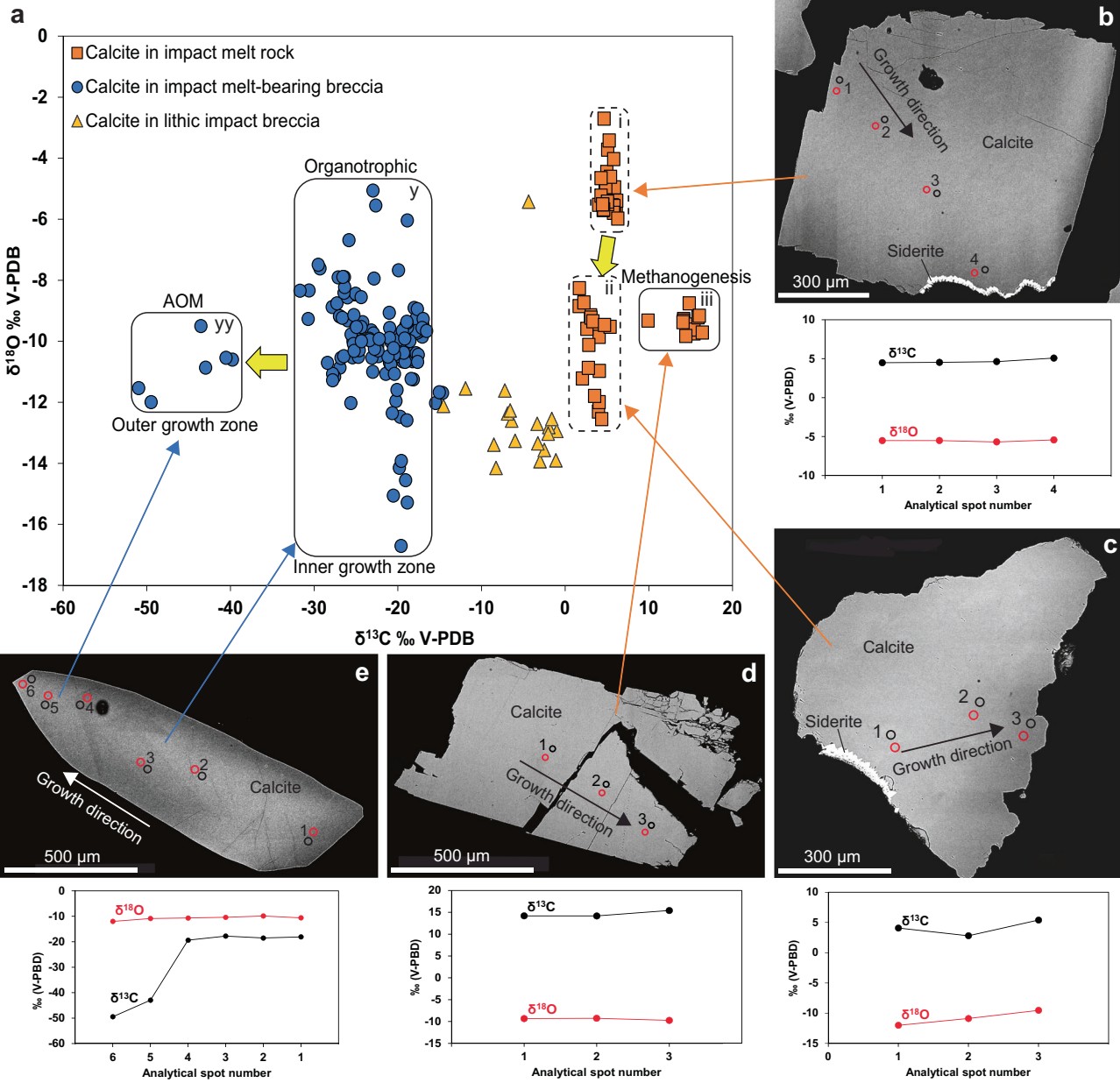

**Fig. 4 | Stable isotope composition of calcite. a** $\delta^{13}C_{calcite}$ vs. $\delta^{18}O_{calcite}$ plot for SIMS analyses of calcite from impact melt rock (orange squares), impact melt-bearing breccia (blue circles) and lithic impact breccia (yellow triangles). Fields for isotopic signatures representing different groups (and processes) are highlighted (i, ii, iii, y, yy). Yellow arrows indicate petrographically identified chronological (as documented by CL, BSE and/or indicated by SIMS data) order from older to younger calcite. **b**–**e** SEM images of polished calcite crystal cross-sections with spot locations for SIMS analyses, corresponding SIMS C and O isotope data. **b** $^{13}C$-enriched calcite from impact melt rock sample LA1;137, plotted in group "i" in (**a**). **c** $^{13}C$-enriched calcite from impact melt rock sample LA1;137, group "ii". **d** $^{13}C$-enriched calcite from impact melt rock sample LA1;30, group "iii". **e** $^{13}C$-depleted calcite from impact melt-bearing breccia sample LA4;152. In the $\delta^{13}C_{calcite}$ vs. $\delta^{18}O_{calcite}$ plot in (**a**), spots 1–4 plots in group "y", whereas spots 5–6 are plotted in group "yy". Each spot represents one SIMS analysis, with errors (1σ) within the size of the symbols.

precipitation varied between different impact rocks and between the veins and vugs. Further, temporal $\delta^{13}C_{calcite}$ and $\delta^{34}S_{pyrite}$ variability within single fractures and vugs suggests episodic precipitation. This requires direct microscale geochronological constraints to distinguish different stages of microbial colonization of the waning stages of the IGH system, as well as subsequent colonization via ambient groundwater circulation. As the fractures and vugs may have hosted communities of different metabolic types, associated with specific isotopic fractionations, mixed signals from different processes can be expected, complicating process determinations.

Microbial sulfate reduction (MSR) leads to low $\delta^{34}S$ values in the produced hydrogen sulfide, as $^{32}S_{sulfate}$ is favored over $^{34}S_{sulfate}$ ref. 32.

Pyrite formation following MSR inherits the sulfur isotopic signature of the precursor hydrogen sulfide[33], making $\delta^{34}S$ in pyrite a common marker for MSR in ancient environments[e.g. 34]. The low minimum $\delta^{34}S$ value of −31.9‰ (Fig. 3a) in the lower impact melt rock section is thus interpreted to reflect MSR. Although the lack of coeval sulfate minerals inhibits direct estimation of the fractionation factor ($\varepsilon^{34} = \delta^{34}S_{sulfate} - \delta^{34}S_{pyrite}$), an initial $\delta^{34}S_{sulfate}$ value can be assumed based on the composition of dissolved sulfate in deep fracture networks of nearby areas of the Fennoscandian Shield. At Olkiluoto, 200 km southwest of Lappajärvi, dissolved sulfate at a depth of c. 375 m exhibits a $\delta^{34}S$ value of c. 25‰[35]. This suggests that an $\varepsilon^{34}$ value of 57‰ is required to reach the minimum $\delta^{34}S_{pyrite}$ values at Lappajärvi. Even if the initial sulfate

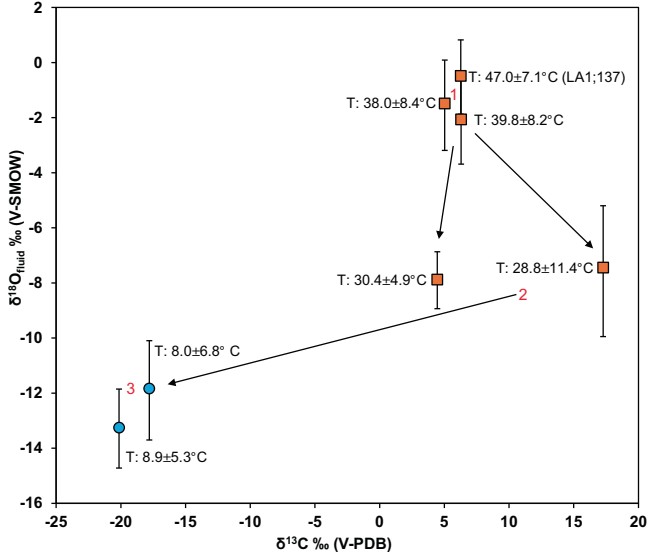

**Fig. 5 | $\delta^{18}O_{fluid}$ based on calcite clumped isotope temperatures and $\delta^{18}O_{calcite}$ vs bulk $\delta^{13}C_{calcite}$ compositions of five calcite samples from impact melt rock (orange squares) and two calcite samples from impact melt-bearing breccia (blue circles), divided into three groups, in chronological order.** The data point for calcite from sample LA1, 137 is marked and represents a micro-drilled sub-sample of the oldest generation of calcite in this sample, whereas the others are bulk samples that may represent several generations. The $\delta^{18}O_{fluid}$ values were calculated using the temperature-dependent fractionation equation $10^3 \ln\alpha_{(calcite-water)} = 2.78 (10^6 \times T^{-2}) - 3.39$ from[31]. The error of $\delta^{18}O_{fluid}$ (Supplementary Data 5) has been propagated from the error in T and $\delta^{18}O_{calcite}$ (Supplementary Data 4). The error of $\delta^{13}C$ (1$\sigma$) is within the size of the symbols.

had a magmatic $\delta^{34}S$ composition of ~0‰, the isotope enrichment (32‰) would still exceed that of abiotic thermochemical sulfate reduction (TSR), which in specific cases reaches up to 22‰[36]. As calcite in the lower impact melt rock has formation temperatures of <47.0 ± 7.1 °C, well below the typical minimum temperature range for TSR (100 − 140 °C)[37], TSR can be excluded as a formation mechanism. Other sources of sulfate, including Cambrian and Ordovician sedimentary rocks that covered the area during impact[38] and seawater at the time of impact, would not yield significantly lower $\varepsilon^{34}$ values, as these had $\delta^{34}S$ values of approximately 15–35‰ and 15–20‰, respectively[39]. In summary, two lines of evidence support MSR: (1) the low $\delta^{34}S$ values of −31.9‰; and (2) the low-temperature formation of coeval calcite. The timing of this MSR process is implied by the calcite U-Pb age of 73.6 ± 2.2 Ma (Fig. 6), indicating that microbial colonization occurred during the hydrothermal circulation in fractures and vesicles of the impact structure.

The substantial $\delta^{34}S$-enrichment of pyrite, of up to 102.2‰ (Fig. 3b), requires further explanation. Under semi-closed system conditions, Rayleigh distillation may cause a continuous increase in $\delta^{34}S$ due to a higher reduction rate of the sulfate pool than its supply by advection and diffusion, leading to a diminished pool of dissolved sulfate[40], which also can be due to TSR[41]. Trends of increasing $\delta^{34}S$ values from core to rim are observed in pyrite within fractures or vugs across all three studied impact lithologies (e.g. LA1;137, Fig. 3a). For example, $\delta^{34}S_{pyrite}$ values in impact melt rock sample LA1;14 increase from −14 to +102‰, suggesting that 96% of the sulfate pool is consumed during MSR when applying Eq. (1)[42]:

$$\delta^{34}S = (\delta^{34}S_{sulfate,0} + 1000)f^{(1/\alpha-1)} - 1000 \quad (1)$$

where $\delta^{34}S$ is the $\delta^{34}S$ value at a given fraction ($f$) of the sulfate reservoir, $\delta^{34}S_{sulfate,0}$ is the initial sulfate value, and $\alpha$ denotes the fractionation factor ($R^{SO4}/R^{H2S}$), where R is the ratio of the heavy to light

isotope. Such heterogeneous isotopic distribution is typical for MSR, but not for TSR for which smaller fractionations result in more homogeneous composition within grains[43].

Microbial methane is typically $^{13}C$-depleted compared to other carbon compounds, leading to residual $^{13}C$-enriched $CO_2$[44]. This $^{13}C$-enriched carbon may be incorporated as bicarbonate into precipitating carbonate minerals, and hence a tracer for methanogenesis as shown by $^{13}C$-rich secondary carbonates in sedimentary basins[45], fractured crystalline rocks in the Fennoscandian shield[46–48] and impact structures[23,27]. At Lappajärvi, the significantly $^{13}C$-enriched calcite (17.3‰) in an impact melt rock vug (Figs. 2c, 4d) is hence proposed to reflect calcite formation following microbial methanogenesis. Correlation of the $\delta^{13}C/\delta^{18}O$, $\delta^{18}O_{fluid}$ and clumped isotope temperature data (Figs. 4a, 5) indicates that it precipitated by the same fluid as the second group (ii) of calcite in LA1;137 at 62.5 ± 3.0 Ma. Taken together, this suggests that – albeit the petrographic observation of $^{13}C$-rich calcite in a vug within impact melt rock –methanogenesis post-dated the impact event by more than 10 million years and, thus, was not related to the IGH-system. This is in accordance with $^{13}C$-enriched calcite in Lockne[27] and Siljan[23] impact structures postdating the impacts by ~100 Myr and by more than 300 Myr, respectively.

During subsurface anaerobic oxidation of methane (AOM) with sulfate reduction,

$$CH_4 + SO_4{}^{2-} = > HCO_3{}^- + HS^- + H_2O \quad (2)$$

The significant $^{13}C$-depletion of the oxidized methane is inherited by precipitated calcite formed from the oxidized carbon species[49]. The $^{13}C$-depletion in the product may be enhanced during AOM by fractionation[50] or diminished due to dilution by other relatively $^{13}C$-rich dissolved carbon species[49] or back-flux reactions at low sulfate concentrations[51]. The low $\delta^{13}C_{calcite}$ values in the lower impact melt-bearing breccia of Lappajärvi (e.g., Fig. 4e: -49.5‰) are thus proposed to reflect calcite formation following AOM. Although the calcite overgrowths with AOM-signature (group yy) are from isolated vugs within the impact melt-bearing breccia, U-Pb dating revealed that the maximum age (43.0 ± 9.6 Ma, Supplementary Fig. 11) of this AOM-generated calcite is younger than the impact event. A relatively young age for this calcite is further supported by the low formation temperature of 8.0 ± 6.8 °C and low $\delta^{18}O_{fluid}$ values, indicating a glacial or meteoric fluid source for the calcite. In comparison, AOM-related $\delta^{13}C_{calcite}$ values of −52.3‰ were reported from the Siljan impact structure[23], postdating the impact by as much as ~340 Myr, showing that impact rocks are prone to later microbial methane cycling.

In summary, the S isotopic data indicates microbial colonization within the late-stages of a slowly cooling IGH system within the Lappajärvi structure (Fig. 7). Immediately following the impact event, this complex crater was filled with an estimated <325 m-thick melt layer (impact melt rock), based on a post-impact erosion estimate of <190 m[52] and drill core observations (Fig. 1). The melt layer experienced initial post-shock temperatures of 1800–2100 °C[53] and overlies impact melt-bearing breccia and lithic (melt-free) impact breccia on top of the fractured target bedrock. The most accurate estimate of the age of the impact event, 77.85 ± 0.78 Ma (Fig. 7), records the closure temperature (> 900 °C) of the U-Pb system in shock-recrystallized zircon immediately after initial quenching of the superheated impact melt[12]. The offset from $^{40}Ar/^{39}Ar$ dating by[13] of impact melt rock of 76.20 ± 0.29 Ma and a K-feldspar melt particle of 75.1 ± 0.36 Ma, reflects the higher isotopic closure temperature for U-Pb compared to the younger $^{40}Ar/^{39}Ar$ ages, which record progressive cooling of different domains of the impact structure[12]. Thermal modeling of the post-impact cooling scenario suggests that temperatures below 250 °C were sustained for up to ~1.6 Myr after the impact event[13], although Kenny et al.[12] estimated slightly faster cooling rates resulting in temperatures >200 °C, locally lasting for more than 1 Myr. The formation temperature

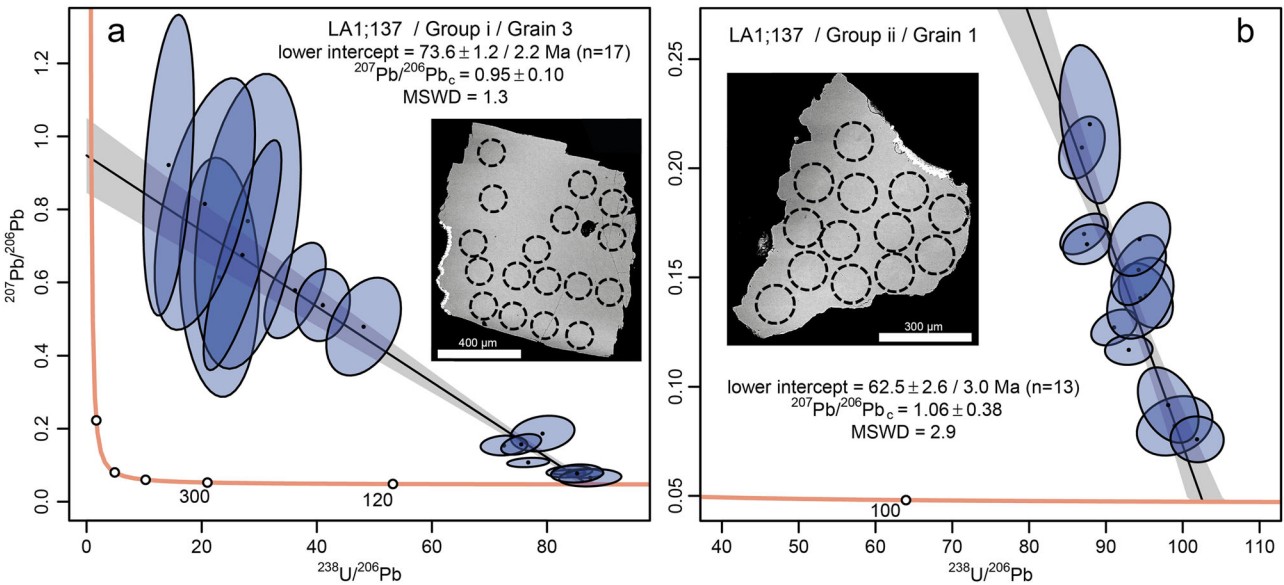

**Fig. 6 | U-Pb dating of calcite sample LA1;137. a** First generation, group i. **b** Second generation, group ii. Quoted ages reflect the lower intercept on a Tera-Wasserburg plot. Uncertainties are quoted at 2σ and reported as ± analytical / analytical + systematic uncertainties. Analytical data are reported in Supplementary Data 6.

$(47.0 \pm 7.1 \,^{\circ}\text{C})$ of fracture-fill calcite with an age of $73.6 \pm 2.2$ Ma, precipitated >1.27 Myr after the impact event (4.25 Myr if not considering any uncertainties), fits the thermal model (Fig. 7) of the decaying Lappajärvi IGH system.

A depth of ~$3.0 \pm 0.5$ km is needed to reach the calcite clumped isotope temperatures of $47.0 \pm 7.1 \,^{\circ}\text{C}$, given the present-day groundwater temperatures[54] and geothermal gradient[55]. Even if assigning increased paleo-groundwater temperatures during the late Cretaceous[56], a depth of ~$2.0 \pm 0.5$ km would be needed (Supplementary Text 1). This is significantly deeper than the estimated depth of calcite formation of 325 m (adding drill core depth to estimated total erosion[52]) and supports the notion that calcite formation was related to a cooling impact-generated hydrothermal system.

The thermal model in Fig. 7 can be placed into the context of a two-stage cooling model of IGH systems[cf. 57], comprising: (1) rapid convection-driven cooling while temperatures are above the boiling point of water through steam production and degassing; and (2) a long period of gradual cooling once convection and steam production cease. In addition, Osinski et al.[58] used fluid inclusions to describe three stages of evolution of the IGH system at Haughton impact structure, which is approximately the same size as Lappajärvi. At Haughton, hydrothermal minerals, such as pyrite, were formed in the second main stage at ~200–80 °C. In the late stage (<80 °C), calcite mineralization occurred at >60 °C within veins and isolated cavities in impact melt rocks and around the margins of the central uplift[58]. The longevity of the hydrothermal system at Lappajärvi, or at least pockets of prolonged fluid flow, compared to the similarly-sized Haughton (~50,000 years) and Ries (~250,000 years), may have several explanations. Given the similar size of these three structures, it is unlikely that the elevated geothermal gradient due to impact-induced uplift or residual heat generated by shock and friction within the central uplift would be that different at Lappajärvi. We suggest that the simplest explanation is due to the predominance of crystalline metamorphic rocks in the target rocks at Lappajärvi. This led to the generation of a thick (currently 145 m but originally likely much thicker) layer of silicate impact melt rock, which is the major heat source for hydrothermal systems in mid-size impact structures[6,7]. In contrast, no coherent bodies of silicate melt rock are known at Haughton and only very small (few m) examples at Ries due to the presence of a thick layer of sedimentary cover rocks at the time of impact. Moreover, the relatively "dry" crystalline target

rocks at Lappajärvi may have led to relatively slow conductive heat transfer from the crater center, and the low permeability of the target rock could have prolonged hydrothermal activity as well (see Supplementary Text 2 for an extended discussion on mechanisms that could explain the longevity of the heat source or pockets within the crater).

At Lappajärvi, vugs (Fig. 8) were formed in the impactites during convection-driven cooling above boiling temperatures through steam generation and degassing, similar to hydrothermal circulation at e.g., the Ries impact crater[57]. An IGH system was generated by the interaction of impact-melted and heated materials with $H_2O$. Eventually, hot fluid circulated in the variably permeable impactites. Based on textural and petrographic evidence in accordance with findings from[13,59], hydrothermal clay minerals, pyrite, and zeolites were among the first mineral coatings to appear at wall of vugs in the impactites in the early IGH stages, but these assemblages could neither be dated nor yield formation temperatures. Pyrite of this generation (Fig. 8b) is proposed to be abiotic and hydrothermal based on the coarse-grained textures and average $\delta^{34}\text{S}$ values of $3.1 \pm 2.1‰$.

The oldest dated record of diagnostic chemofossils at Lappajärvi was MSR-induced pyrite intergrown with calcite formed at $73.6 \pm 2.2$ Ma (Fig. 8c) in a fracture in the impact melt rock a few meters above the underlying impact melt-bearing breccia. The Chicxulub, Ries and Sudbury impact structures show similar increased alteration close to lithological contacts, owing to that the contacts permit more fluid flow[60, and references therein]. At this stage, the IGH system was still slowly cooling down, as indicated by the formation temperature $(47 \pm 7.1 \,^{\circ}\text{C})$ of the calcite. At $62.5 \pm 3.0$ Ma, following further cooling to $28.8 \pm 11.4 \,^{\circ}\text{C}$ and erosion, infiltration of fluids with a meteoric water end-member—based on $\delta^{18}\text{O}$ values—featured microbial methanogenesis in the upper impact melt rock vugs and precipitation of calcite in fractures in the lower impact melt rock (Fig. 8d). At $43 \pm 9.6$ Ma, following additional cooling to $8.0 \pm 6.8 \,^{\circ}\text{C}$ and erosion, organotrophic microbial colonization of vugs in the impact melt-bearing breccia took place at a wide range of depths revealed by $^{13}\text{C}$-depleted calcite (Fig. 8e). At this stage, the cooling of the IGH system had long-since ceased and the low-T fluid by which the organotrophic calcite and outermost overgrowth of AOM-related calcite (Fig. 8f) precipitated, was probably meteoric, based on $\delta^{18}\text{O}$ values. These mineral-sealed primary vugs in the impact melt rocks were sampled to investigate processes related to the impact event based on the petrographic

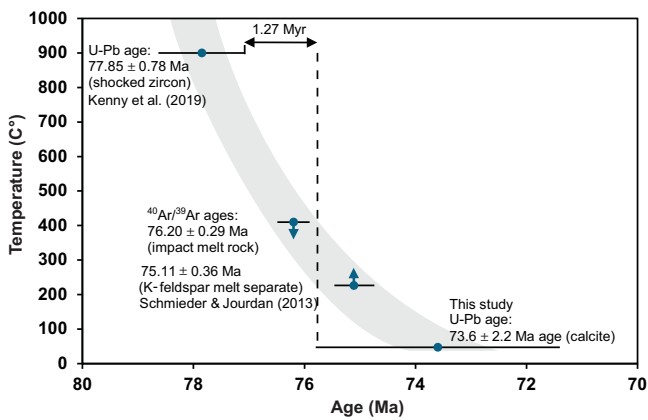

**Fig. 7 | Radioisotopic ages for Lappajärvi rock materials used as a thermal model of a waning IGH system.** A comparison of (i) the U-Pb concordia age of shocked zircon (77.85 ± 0.78 Ma; *n* = 8 measurements)[12] with (ii) the previous 'best estimate' impact age (76.20 ± 0.29 Ma; *n* = 7 samples), calculated from the $^{40}Ar/^{39}Ar$ age of impact melt rock and a syn-melt rock K-feldspar melt particle separate[13], (iii) $^{40}Ar/^{39}Ar$ age of the youngest K-feldspar melt particle separate (75.11 ± 0.36 Ma; *n* = 1 sample)[13], and (iv) formation temperature (*n* = 6 replicates) and U-Pb concordia age of calcite (73.6 ± 2.2 Ma; n = 17 measurements) from the present study. Grain-size dependent argon diffusion parameters determined for various domain sizes of K-feldspars define the apparent closure temperatures of -230–410 °C[13] given in the diagram, and the ages are set to represent this span. Hence, the arrows on the symbols which indicate that they represent maximum and minimum temperatures. More thorough discussions of the $^{40}Ar/^{39}Ar$ ages and related diffusion modeling are in ref. 13. All uncertainties, including those related to the decay constants, are included. For (iv), the error of temperature (°C) for calcite is within the symbol. The gray area represents a polynomial fit trendline.

observations. However, our work shows that most vugs in the impact melt-bearing rock units were colonized more than ten million years after the vugs were formed and hydrothermal circulation had ceased, demonstrating the importance of applying direct geochronology of the biosignature-hosting minerals, and not merely relying on petrographic evidence. With this in mind, it would be important to (re)consider previous studies from the Haughton, Rochechouart and Chicxulub impact structures—that suggested MSR-induced sulfides to be proof of microbial colonization of impact structures within the lifetime of the IGH systems[21,22,24]—as these studies did not provide geochronological data, and temperature constraints were presented from veins at Haughton only[21,58].

This study provides coupled isotopic-temperature-geochronology support for microbial colonization in the waning stages of an impact-generated hydrothermal (IGH) system, at a stage when temperatures had dropped to levels where life can sustain. These findings corroborate previous claims[12,13] that IGH circulation can be sustained for an extended period, even in comparatively small complex impact structures such as Lappajärvi. As impact events have been suggested to potentially provide a mechanism to generate habitable planets, satellites, and even asteroids throughout and beyond the Solar System[2], the present study gives important insights into microbial activity in IGH systems and their associated craters[1]. The microbial colonization of the Lappajärvi impact structure may thus serve as an analog for the emergence of life on Early Earth and possibly Mars. The analytical protocol is suitable for studies of microbial colonization of other impact structures on Earth, on other planetary bodies, particularly for future sample return campaigns.

## Methods
### Sample materials and site
The 23 km diameter Lappajärvi impact structure in central-western Finland is a well-preserved, albeit moderately eroded impact

structure[61]. A concentric arrangement of impact-produced crater-fill deposits remains underneath Quaternary sediments (Fig. 1). Impact melt rock currently constitutes a 145 m thick coherent layer. The upper 24 m and lower 26 m of the impact melt rock have a crystalline matrix in the form of recrystallized glass, whereas the intermediate impact melt rock has a glassy (perlitic) matrix[38references therein]. The impact melt rock is relatively homogeneous and can be reproduced by a mixture of the Paleoproterozoic (ca. 1.90–1.87 Ga) target rock: 76 vol.% mica schist, 11% granite pegmatite, and 13% amphibolite[62]. The impact melt rock is both underlain and surrounded by impact melt-bearing breccia and lithic impact breccia. In the impact melt rock, porosity is around 1%, whereas in impact melt-bearing breccia and lithic impact porosity is as high as 20%, which is significantly higher than in unweathered crystalline rocks, in which porosity is typically less than 0.5%[63]. Gradually, the impact-produced crater-fill deposits proceed downward and outward from the central part of the crater into autochthonous impact breccia, and finally to the fractured bedrock[29]. The outer margin of the crater is rimmed by a megablock zone[38] of autochthonous breccia in which a circular depression (ring graben) is filled with allochthonous Mesoproterozoic sedimentary rocks[29], which locally overlaid the crystalline bedrock at the time of the impact, at a thickness of less than 200 m[38]. Samples were collected from two cored boreholes (LA1 and LA4 in Fig. 1) drilled into the impact structure.

Impact melt rock in drill core LA1 was sampled in intervals with abundant mineral-filled cm-sized vugs at 3–10 m and 144–145 m depth (just above the contact to impact melt-bearing breccia) and fractures at 110–140 m depth. The 10–110 m interval has few vugs and sealed fractures, but a calcedony interval occurs (sampled). Impact melt-bearing breccia in LA4 was sampled in intervals with abundant mineral-filled mm- to cm-sized vugs at 23–35 m (high proportion of vugs), at 35–66 m (moderate proportion of vugs), at 66–76 m (fragmented and heavily weathered interval, high proportion of vugs) at 76–82 m (high proportion of vugs) and at 142–152 m (moderate proportion of vugs). Lithic impact breccia in LA4 was sampled in intervals with abundant mineral-filled cm-sized vugs at 83–90 m (very high proportion of vugs), at 90–95 m (heavily weathered and partially dissolved interval, high proportion of vugs), at 95–106 m (heavily weathered interval, high proportion of vugs) and at 106–148 m (moderate proportion of vugs).

### Sample preparation
Thirty-three rock core intervals containing open, partly sealed or completely sealed fractures and vugs were targeted and sampled for secondary coating minerals, focusing on calcite and pyrite, for stable isotopes, radioisotopic dating, X-ray diffraction of clay minerals (method in Supplementary Text 3, data in Supplementary Data 8) and organic molecules (too contaminated to be included, see Supplementary Text 4). Eight of the intervals are from the 218 m long and 42 mm diameter (plunge: 80°, trend: 030°, year: 1988)[38] drill core K231388R301 (here denoted LA1). The remaining 25 intervals are from the 275 m long and 55 mm diameter (plunge: 80°, trend 030°, year 1990)[38] drill core K231390R304 (LA4). The crystals were characterized under a stereomicroscope and identified directly on the uncoated surfaces of vugs, veins and fractures at low-vacuum mode and for carbon-coated thin sections at high-vacuum mode using a FEI QUANTA FEG 650 Scanning Electron Microscope (SEM) equipped with an integrated energy dispersive spectroscopy (EDS, Oxford T-Max 80 detector). The acceleration voltage was 20 kV, and the instrument was calibrated with a cobalt standard.

### Secondary Ion Mass Spectrometry (SIMS)
Following sample characterization and mineral identification, single crystals or aggregates were separated from fractures and vugs with a chisel and/or tweezers, under a stereomicroscope, and mounted in epoxy. The epoxy-mounted grains were polished to expose crystal

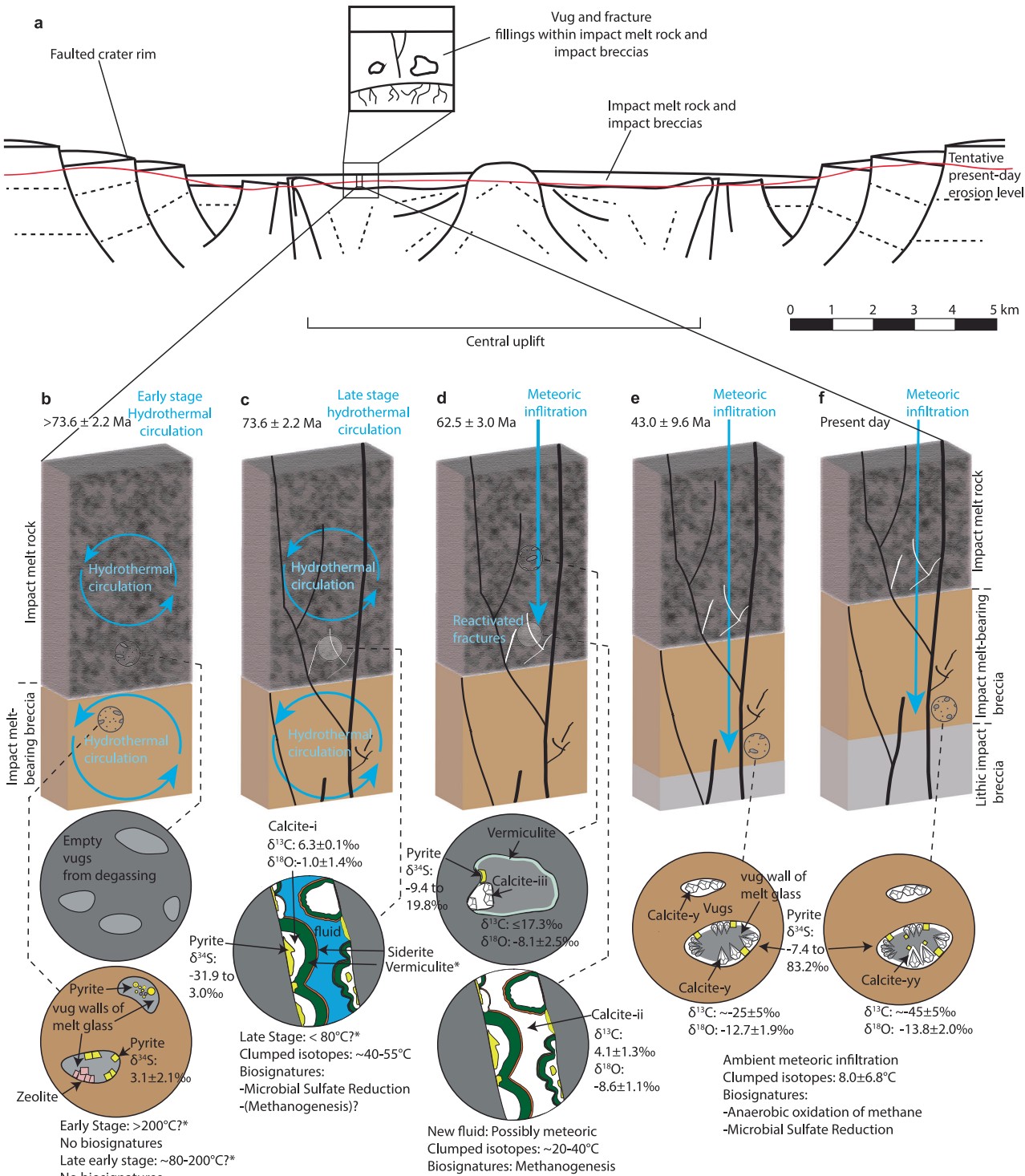

**Fig. 8 | Schematic post-impact evolution and microbial colonization of impactites in the Lappajärvi impact structure. a** A schematic cross-section showing the presence of vugs and fractures within crater-fill impactites (redrawn from[58]). Viewed from the southwest, the tentative present-day erosion level is interpreted from[29,38]. Key stages (**b**–**f**) are shown with a record of biosignatures, environmental and geochronological constraints, including formation temperatures of calcite, $\delta^{13}C_{calcite}$, $\delta^{18}O_{fluid}$, and $\delta^{34}S_{pyrite}$ in vugs and veins, mineral precipitation events, biosignatures for different microbial metabolisms, and fluid types. An erosion rate of -2.5 m /Myr[52] accounts for the exhumation in the model. Temperatures (*) for stages of the waning IGH system in (**b**) and (**c**) are estimates based on[58]. In (**f**), $\delta^{18}O_{fluid}$ was calculated from $\delta^{18}O_{calcite}$ and temperature for the whole crystals, even though temperature is not known specifically for the overgrowth.

cross-sections and characterized in SEM once again to identify crystal zonation, cracks, mineral inclusions and impurities, and to optimize SIMS spot placement. Intra-crystal SIMS-analysis (10–μm lateral beam dimension, 1–2 μm depth dimension) of carbon, oxygen and sulfur isotopes was performed on a CAMECA IMS1280 SIMS at NordSIMS, Swedish Museum of Natural History, Stockholm, Sweden. Thematic spatial correlation of different techniques used for the various mineral materials, are detailed in Supplementary Figs. 3–5.

### SIMS-Pyrite

Analytical transects were made from core to rim within 53 pyrite crystals from 24 drill core samples. In total, 344 analyses were made for $\delta^{34}S$. Sulfur was sputtered using a $^{133}Cs^+$ primary beam with 20 kV incident energy (10 kV primary, −10 kV secondary) and a primary beam current of ~1.5 nA. A normal incidence electron gun was used for charge compensation. Analyses were performed in automated sequences, with each analysis comprising a 70 s pre-sputter to remove the gold coating over a rastered $15 \times 15\,\mu m$ area, centering of the secondary beam in the field aperture to correct for small variations in surface relief and data acquisition in sixteen four-second integration cycles. The magnetic field was locked at the beginning of the session using an NMR field sensor. Secondary ion signals for $^{32}S$ and $^{34}S$ were detected simultaneously using two Faraday detectors with a common mass resolution of 4860 (M/ΔM). Results are reported as‰ $\delta^{34}S$ based on the Canon Diablo Troilite (V-CDT)-standard value[64]. Full data, including reference materials, are in Supplementary Data 1.

### SIMS-Calcite

Analytical transects were made in 56 calcite crystals collected from 15 drill core samples. In total, 449 analyses (settings follow[46]) were made, 220 for $\delta^{13}C$ and 229 for $\delta^{18}O$. Influence of organic matter and inclusions of other minerals was avoided by careful spot placement to areas in the crystals without micro-fractures or inclusions. The uncertainty associated with potential organic inclusions and matrix composition is therefore considered to be insignificant compared to the isotopic variations. Calcite results are reported as‰ $\delta^{13}C$ and $\delta^{18}O$ based on the Pee Dee Belemnite (V-PDB)-standard value. Analytical sessions were carried out, running blocks of six unknowns bracketed by two standards. Spot transects were made from core to rim within the crystals (full data, including values of reference materials in Supplementary Data 2). Corresponding analytical spots for C and O isotopes were closely placed within the crystals and analyzed at separate sessions. Isotope data were normalized using calcite reference material S0161, a granulite facies marble in the Adirondack Mountains, courtesy of R.A. Stern (Univ. of Alberta). The values used for IMF correction were determined by conventional stable isotope mass spectrometry at Stockholm University on ten separate pieces, yielding $\delta^{13}C = -0.22 \pm 0.11‰$ V-PDB (1 std. dev.) and $\delta^{18}O = -5.62 \pm 0.11‰$ V-PDB (1 std. dev.). Precision was $\delta^{18}O$: ± 0.2−0.3‰ and $\delta^{13}C$: ± 0.4−0.5‰. Calcite trace element concentrations may influence SIMS matrix effects, at least for $\delta^{18}O^{ref.65}$, but since 1) the 3−4 mass% of Fe+Mn measured with EDS in the calcite at Lappajärvi (Supplementary Fig. 3) may only influence the $\delta^{18}O$ by a maximum of ~1‰ when applying a correction of Rollion-Bard and Marin-Carbonne[65], and 2) the clumped isotope measurements and SIMS measurements generally overlap for same samples, we have not applied any correction based on trace element content to the SIMS data.

### Carbonate clumped isotope analysis

About 2 mg of powdered carbonate was reacted with dehydrated phosphoric acid under vacuum at 70 °C. The conventional isotope ratio measurement relative to V-PDB was calibrated based on repeated measurements of International Atomic Energy Agency (IAEA) reference materials. Clumped isotope values (Δ638) are reported in the Intercarb-Carbon Dioxide Equilibrium Scale (I-CDES) reference frame defined by ETH 1−4 carbonate standards and Carbon Dioxide Equilibrium Scale (CDES) reference frame defined by water-equilibrated gases and heated gases[66,67]. The Δ638 notation refers to measurements using a tunable infrared laser differential spectrometer (TILDAS), performed at the Environmental Isotope Laboratory at the University of Arizona. The precisions for these measurements are ± 0.021‰ for Δ638, ± 0.04‰ for $\delta^{18}O_{carb}$, and ± 0.03‰ for $\delta^{13}C_{carb}$ (pooled reproducibility[68], 1σ), respectively. The laser-based Δ638−temperature relationship for carbonates at relatively low temperatures is based on 44 synthetic calcites that equilibrated at temperatures from 6 to 70 °C[69]:

$$\Delta638_{CDES} = (0.0405 \pm 0.0006) \times (10^6/T^2) + (0.1822 \pm 0.0061),\ R^2 = 0.985 \quad (3)$$

Temperature estimates using laser-based Δ638 and mass spectrometry-based Δ47 measurements have been shown to be statistically equivalent[69]. Equation (3) is within error of recent mass spectrometry-based Δ47-temperature relationships and correctly predicted precipitation temperatures for a suite of 17 natural carbonates for both calcite and aragonite[69]. Complete details of this method and the calibration of the Δ638−temperature relationships are described in[70,71]. For sample LA1;137, a mini drill was used to separate the oldest generation (group i) of calcite for clumped isotope analysis. All clumped isotope samples were analyzed for three to six replicates. Each replicate measurement is based on four comparisons (sub-cycles) of the sample and reference gas. We exclude replicates with fewer than two sub-cycles to calculate the mean value of each sample. The reported clumped isotope values were averaged from the three to six accepted replicates. Supplementary Data 3 and 4 provide full clumped isotope data, including excluded replicates.

### LA-ICP-MS U-Pb Geochronology

U-Pb geochronology via the in situ LA-ICP-MS method was conducted at the Geochronology & Tracers Facility, British Geological Survey (Nottingham, UK). The method utilizes an Elemental Scientific Lasers ImageGeo excimer laser ablation system, coupled to a Nu Instruments Attom single-collector ICP-MS. The method is previously described[71] and involves a standard-sample bracketing with normalization to either 91500 zircon[72] or NIST 614 silicate glass for Pb-Pb ratios[73] and WC-1 carbonate for U-Pb ratios[74]. The laser parameters comprised either a 60 or 120 μm static spot, fired at 10 Hz, with a ~ 4 J/cm$^2$ fluence, for 30 s of ablation. No common lead correction is made; ages are determined by Model 1 regression and the lower intercept on a Tera-Wasserburg plot (using IsoplotR[75]). Duff Brown, a carbonate previously measured by Isotope Dilution mass spectrometry, was used as a validation and yielded pooled ages of 62.6 ± 0.78 Ma (unanchored regression) or 66.76 ± 0.72 Ma (anchored to $^{207}Pb/^{206}Pb_c = 0.7385$), overlapping the published age of 64.04 ± 0.67 Ma[76]. All ages are plotted and quoted at 2σ and include propagation of overdispersion and systematic uncertainties according to recommended guidelines[72]. Data are screened for very low Pb and U counts below detection ( < 20 cps), and anomalously high ratio uncertainties ( > 50%). Eight samples were analyzed in total, with only three providing robust age constraints. Analyses were conducted on individual crystals that were previously used for in situ stable isotope analysis via SIMS. Full analytical data are listed in Supplementary Data 6; analytical conditions in Supplementary Data 7, and further details on the analytical sessions are given in Supplementary Text 5.

### Reporting summary

Further information on research design is available in the Nature Portfolio Reporting Summary linked to this article.

## Data availability

All relevant data are included in the Supplementary Material of this article and stored publicly available in the archive at Swedish National Data Service (doris.snd.gu.se) and available here: https://doi.org/10.5878/mmsk-2e30.

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

## Acknowledgments

This work was supported by Swedish research council (contract 2021-04365), Formas (contract 2020-01577), J. Gust. Richert Foundation (contract 2023-00850), the Crafoord Foundation (contract 20230005), and the National Science Foundation (contract 2120733), all to HD. We thank Manuel Reinhardt for biomarker analysis and sample preparation, the Geological Survey of Finland, Loppi drill core facility with crew for sampling assistance, Teemu Öhman and Gavin Kenny for discussions, Kerstin Lindén for assistance with SIMS sample preparation, and the ongoing collaborative effort of the Biology of Biosignature Detection group from the Keck Institute for Space Studies. This is NordSIMS publication #799.

## Author contributions

J.G. carried out sampling, sample preparation, SEM, and SIMS analyses, did the conceptual modeling and drafted the paper. G.O.: sampling, and assistance with conceptual modeling and drafting, N.R.: U-Pb geochronology and data reduction. J.Q. and Z.W.: clumped isotope measurements and data reduction. M.W. and H.J. SIMS analyses and data reduction, A.K. SEM and XRD, S.H. provided samples and assistance

during sampling. H.D. initiated the study, carried out sampling, SIMS analyses, conceptual modeling and provided funding. All authors edited the final draft.

## Funding

## Competing interests
The authors declare no competing interests.
