## [Peer Review file · Nature Communications]

Deep microbial colonization during impact-generated hydrothermal circulation at the Lappajärvi impact structure, Finland

Corresponding Author: Dr Henrik Drake

Version 0:

Reviewer comments:

Reviewer #1

(Remarks to the Author)

Dear Dr Rosch,

thank you for choosing me as a reviewer of the manuscript "Deep microbial colonization during impact-generated hydrothermal circulation at the Lappajärvi impact structure, Finland", submitted to Nature Communications by Drake et al.

Overall, this is a well-written, but arguably quite technical, manuscript. Its actual core message, which is to highlight medium-size (and large) terrestrial impact structures as long-lasting hotspots for the emergence and evolution of microbial life in a hydrothermal setting as the crater cools down to ambient conditions, could be conveyed a little more clearly especially in the abstract. The study presents some fascinating (and, as far as I can tell, reliable) U-Pb age results gained from calcite that suggest the crater cooling process lasts, in fact, several million years even in the case of a ~20 km crater – which may be surprising to some (but less so to others). This would make Lappajärvi the single most intensely studied impact crater in terms of geochronology, using different geochronometers and dating minerals, regarding the crater cooling question. It also makes me wonder – if Lappajärvi stays warm for this long – how long will much larger, Chicxulub-size impact craters, take to fully cool down?

If the authors manage to set out their core message and its relevance in a somewhat more striking way, I think publication in Nat. Comm. will be justified. However, looking at the present manuscript I'd rather recommend a more specialized journal, e.g., "Astrobiology", "Earth and Planetary Science Letters", or "Geochimica et Cosmochimica Acta". I also encourage the authors to add a brief discussion of potential mechanisms that could explain the surprising longevity of the heat source or pockets within the crater.

For the manuscript itself (regardless where it will be published) I recommend minor revision.

Specific comments:

Abstract, line 24: I'd write "10 Myr later", since this describes a time span, and not an age. (I acknowledge some journals only accept Ga/Ma/ka as geologic time units and sometimes change those within the course of the proofing and typesetting process.)

Line 35: Two other suitable references would be Schmieder & Kring (2020, Astrobiology, with a focus on impact ages) and Kenkmann (2021, MAPS).

L 52: ~1 to 1.6 Myr after... see comment above. "1.6 Ma" is a point in time within the Pleistocene.

L 200/201: Here and in the following lines, the authors report a number of "temperature reconstructions" and choose to label them "temperatures". This sounds a bit like isotope jargon slang and might be misleading to some readers. Perhaps add "... yielded temperature reconstructions (henceforth 'temperatures'), ranging from...". Alternatively, those temperatures could be called paleotemperatures, mineralization temperatures, or simply reconstructed temperatures.

L 326-328: There is something wrong with the sentence "Thermal modelling suggests that cooling below 250 °C took place up to ~1.6 Ma after the impact event¹², although¹¹ estimated slightly faster cooling of >200°C, locally lasting for >1 Myr." Are the authors talking about post-impact cooling rates? Also see comment above regarding Ma vs. Myr.

L 333: The authors touch on the geothermal gradient here. This is an interesting aspect. The authors should, at least briefly, discuss the anatomy and role of the central uplift at Lappajärvi, which is not prominent morphologically/topographically as it is concealed by the impact melt sheet. Nevertheless, an impact crater the size of Lappajärvi should be characterized by some ~2.x km of structural uplift within the crater basement at its center. This uplift may also affect the original geothermal stratification within the target rock, i.e., leading to an upward 'heat bulge' (compare Naumov, 2005; Schmieder & Jourdan, 2013). Would there be any measurable effect of a ~2 km uplift of the geothermal isotherm pattern at Lappajärvi, and could this help explain the ~73 Ma low-temperature 'warm oasis' that we see in the calcite U/Pb results? Is Lappajärvi associated with a resolvable geothermal anomaly (compared to the surrounding bedrock) in geophysical maps today?

L 343: check "a three-stages of evolution"

Fig. 2d: are the two roundish pyrite grains near the bottom of the right image framboidal pyrite? Or just round crystalline grains? Just as a remark: there are also Fe-Ni-rich metal and sulfide particles in the Lappajärvi melt lithologies that may contain remnants of the impacting projectile (see Fregerslev 1976). Maybe this is worth mentioning.

Supplementary file and main text: If possible and data exist, the authors could elaborate in a little more detail in which way (if at all) the calcite isotope results are influenced by any internal mineral zonation, e.g., with Fe-, Mn-, and/or Mg-rich zones within the calcite domains (the authors already mention ankerite in the text). At Chicxulub we see an interesting zonation pattern both in SEM-BEI and CL between a Mn-poor and a Mn-rich variety in post-impact hydrothermal sparry carbonate.

Table S6, Excel file: It might be useful to see, in addition to the raw/corrected U/Pb data, individual calculated spot ages with errors, as well.

Apparent age values should be rounded appropriately (I don't think it is useful to report a number as "0.032 +/- 567.610 Ma").

*** End of review ***

Kind regards

Reviewer #2

(Remarks to the Author)

Congratulations on your excellent contribution to enhancing our understanding of when deep subsurface microbial life was able to recolonize impact-deformed geological formations and their role in deep subsurface elemental cycling (i.e., microbial sulfate reduction and anaerobic methane oxidation).

I have only one comment: Many studies have shown that these microbial processes are ongoing even in nutrient-deprived extreme deep biosphere environments, such as described here. A recent survey from Quraish et al. (2024) in *Geobiology* (not cited) showed that extant microbial communities associated with mineral veins and contact zones between breccia and granites, etc., in the granitic basement below the Chicxulub impact crater contain the genetic machinery for chemolithoautotrophic carbon fixation and organoheterotrophy plus cycling of S, Mn, and Fe.

To what extent can the precise dating, identification of minerals, and isotopic analysis on a microscale exclude the possibility that modern activities caused bias of the isotopic results? For example, extant chemoautotrophic activities could result in the deposition of microbial carbonates enriched in $\delta^{13}\text{C}$ (overshadowing $\delta^{13}\text{C}$ depleted signals indicative of the onset and extent of methane cycling?). I suggest the authors discuss this so that microbiologists interested in this paper know that you have considered it even if it is perhaps a no-brainer for specialist readers that the microscale analysis can rule out that some of the analyzed material was deposited more recently and has no impact on the data or outcomes of the study.

Reviewer #3

(Remarks to the Author)

This is a very interesting paper that I thoroughly enjoyed reading. I appreciate the coordination between stable isotope and radiometric analyses to understand the timing of microbial colonization as it is linked (or not linked) to impact-generated hydrothermal systems. This specific type of work in the field of terrestrial impact cratering, and particularly at the intersection of astrobiology, is sorely needed and is highly relevant as we continue to explore craters on Earth and the ancient Martian surface (and potentially subsurface, in the future!)

The work presented here addresses a key question related to impact-generated hydrothermal (IGH) systems which is timing, as it relates to post-impact colonization by microbial communities. Previous studies on sulfide mineral isotopes, for example, in impact craters have identified biosignatures in the form of highly reduced (isotopically light) $\delta^{34}\text{S}$, implying a biogenic rather than (abiotic) thermochemical sulfate reduction origin, but the timing of this activity has remained ambiguous and often linked directly to the hydrothermal system without more definitive (e.g., radiogenic isotope) constraints.

I have a few comments on general observations of the manuscript as well as line-by-line suggestions, below.

Sections of the methods and results related to mineral identification and lithologic context – and by extension the corresponding isotopic analyses - are unclear and would benefit from better organization and clarification. I find it difficult to interpret the isotope data without a better understanding of the context and what exactly has been analyzed. Are many of these generational relationships implied based solely on BSE images of rough sample surfaces? Is there supporting chemical data (e.g., something that clearly shows different generations of calcite) or possibly thin section images to support these implied relationships? If that data is available it would be immensely helpful for the reader and solidify some of these interpretations. I understand this paper represents a lot of coordination between in-situ and bulk analyses, and care should be taken to make it clear what has been analyzed, actually identified and what it represents.

- For example, to examine the identification of a “serpentine-group clay” (Lines 122, 126, 144) I would expect XRD data: I see one XRD pattern - supplemental Fig. 9 - with no labeled peaks or mineral ID's, and S8_XRD representing only one analysis, with a list of potential library matches, but I see no labeled clay (or other mineral) d-spacings indicative of serpentine, vermiculite or corrensite 001's, 060's. Nor do the methods describe the careful analyses to identify corrensite, serpentine or vermiculite, a high charge smectite, were undertaken. The caption also states this was the clay coating on the vug, in the hand sample, but it's unclear how this context was determined - if this is a bulk, crushed sample of a melt rock it could have been a secondary clay in the matrix, or something else as clays can form from altering many of the primary aluminosilicate phases under hydrothermal conditions.

- As another example, I see lots of beautiful photographs and BSE images of rough sample surfaces but no thin sections or chemical (EDS) data to support these interpretations. If this is available (Lines 461-464 suggest EDS analyses should exist), please add a few figures (e.g., chemical maps, spot analyses, something to help place the isotopic analyses and mineral, lithologic relationships into context).

- As a final example, the context for many of these minerals used for isotopic analysis is unclear. E.g., It appears there are more than one occurrences of pyrite in these samples (Fig. 2a vs 2d). Where are the pyrite crystals shown in Supplemental Fig. 3 (and by extension, the longer list of sample ID's in the excel sheet) from, other than “the melt rock” or “the lithic impact breccia?” Are these fracture coatings or vug-fillings (e.g., Fig. 2)? Are they potentially hand-chiseled out (Lines 466-468) from the melt or breccia matrix – some of these are 100 microns or less, as low as 10 microns (Line 134) in size? These relationships really matter when it comes to interpreting their isotopes, particularly as there is such an unusually large range of data here. I have no doubt that with the amount of work performed here, the data confirming these mineral identifications and lithologic, contextual relationships is available to show the reader?

This is exciting work and I hope the authors can take the steps to organize, present, coordinate and clarify their methods and results, especially the implied mineral identifications, relationships and context. This is a really interesting study and I would look forward to seeing some of this information clarified to better piece together the story presented here!

Line by line comments:

Lines 49-52: As written it is unclear whether this statement implies that hydrothermal circulation and cooling below 250 degrees began at 1-1.6 Ma or lasted this long. Please clarify.

Line 53: Minor typo: “...modeling have predicted...” to “...modeling has predicted...”

Lines 75-78: I would argue here that Chicxulub is an exception - those framboids are intergrown with the dachiardite and analcime - these zeolites are high(er) temperature hydrothermal minerals, supporting these LIKELY formed during a period of elevated temperatures. The analcime samples in particular - the images in Kring et al. (2021), and the mineralogical, in-situ context of those crystals - it's hard to make the argument that the framboids formed much later, during diagenesis/far below and outside IGH conditions.

Lines 80-85: Very minor comment – I agree with the overall statement – that there is a need for timing constraints with the biosignature examples described in the previous section, but the biosignatures described in this specific section are very different from the ones listed in the previous paragraph, which is a tad misleading (e.g., 34S of sulfides attributed to bacterial sulfate reduction at elevated temperatures). I would also suggest adding a couple words here that indicate these biosignatures were initially attributed directly to impact hydrothermal activity. When I first read this I thought “well yeah these old rocks are probably covered in some sort of fungus” but went to the reference (and was surprised) to see someone initially claimed these were hydrothermal system-hosted fungi! Similar comment for calcite, which can easily be reset and even initially form at surface diagenetic conditions.

Lines 97-98: The comment on methane seepage is a bit out of place. I understand it's relevance but I would suggest removing this entirely or elaborating on how this is tied to the rest of the sentence and section (e.g., are the authors implying the methane seepage was directly linked to IGH system-hosted microbes?) I would err on the side of caution here, as the source of Martian methane detections are a highly contested topic, and very unlikely to be IGH system related.

Line 131-133: “Clay minerals, zeolite and pyrite were among the first mineral coatings to appear intergrown with impact glass (Supplementary Fig. 1a,c)” Clay minerals and pyrite are not the first secondary minerals to form in these systems – they are often later stage, especially clays. What is the evidence for this, based only on the BSE images? I also see no clays labeled or identified here. Is there EDS data to go with these images?

Line 132: Which zeolite? Or change to zeolites.

Line 140, Figure 2: Is there a difference in composition between these different types of calcite?

Line 143: Where is the data indicating this is ankerite? I see no labeled XRD patterns or in-situ EDS data on thin sections?

Lines 149-150: What pyrite is being analyzing here exactly?

Lines 163-170, Figure 3: what is the context for these pyrite grains (e.g., matrix, fracture or vug coating?) Additionally, what is the other mineral in 3b?

Lines 192-193: How are these petrographically identified as chronologically older? Based on BSE images from rough sample surfaces? Are there thin sections (and corresponding EDS analyses) of these samples?

Lines 271-275: These are very unusual and heavy values for sulfides – a very interesting and exciting find! I see examples of these isotopically heavy sulfides in Fig. 3b and c – any chance you also have corresponding EDS data on these? Or the other mineral intergrown with the pyrite in 3b. I'm curious to know if there's a chemical difference or zoning – anything – that might help interpret these unexpected results.

Lines 452-464: I see methods listed here for low vacuum analyses on rough sample surfaces. Is this what the chronological relationships and mineral identification, context is based on? Other than a single sample bulk XRD analysis and the second set of "SEM analysis" of hand-chiseled and epoxy mounted minerals listed 466-471? Once the samples are hand chiseled you lose a lot of contextual information. I'm curious why no thin section petrographic work was completed. If it was (e.g., EDS mapping at high vacuum, or spot analyses and imaging to understand the in-situ mineralogical, chemical context), please add it to the manuscript.

Supplemental Fig. 1,2: I see several BSE images showing beautiful mineral surfaces, and vugs but no chemical (e.g., EDS, WDS) or crystallographic data (e.g. XRD, outside the unlabeled Supp. Fig. 9).

Supplemental Fig. 9: Please label mineral peaks here (e.g., mineral ID and their respective, indicative 001, 060 d-spacings). How was vermiculite characterized? Clay minerals are notoriously difficult to identify – vermiculite is a high-charge smectite. Is there additional information that shows this is indeed vermiculite? What about the other clays listed in the manuscript – serpentine and corrensite?

Version 1:

Reviewer comments:

Reviewer #1

(Remarks to the Author)

Dear Editor,

having reviewed the second round of this manuscript submission I think the paper is now good to go, with a few final suggestions listed below.

L 29. I'd write „..., which may have...”

L66/L67: 2 x “have been reported”

L72: ...are also...

L97: Yes, but there may be other contexts, e.g., hydrothermal mineral assemblages and sequences with traces of colonization linked to specific minerals using purely petrographic methods, providing a relative timeline of colonization.

L121: maybe remove “unshocked” and replace by “fractured” – this unit was presumably also shocked, but at low levels not immediately recognizable in terms of shock metamorphism.

General: Please proofread a final time for the filling word ‘also’ – there is (locally) an overabundance. Perhaps replace by “Moreover,” or “as well” where appropriate.

L195: 4 cm-long or 4 cm-diameter vug?

L251: comma after parentheses

L300: are consumed?

L491: Sentences should not start with a number. Perhaps write out “Thirty-three...”

Kind regards,
Martin Schmieder

*** End of review ***

Reviewer #2

(Remarks to the Author)

Deeply fractured rocks within meteorite impact structures have been hypothesised as hotspots for microbial colonisation on Earth and potentially on extraterrestrial planetary bodies. Biosignatures have been identified from a limited number of impact structures (notably the Chicxulub impact crater) and have offered valuable evidence of microbial activity following impact events. However, the timing of microbial colonisation in relation to these impact events remains uncertain. Consequently, it is possible that the biosignatures previously reported from impact structures are unrelated to the impact crater formation and instead reflect outcomes of long-term recolonisation of the site.

To test this possibility, this study employed microscale stable isotope biosignature detection, combined with radioisotopic dating, to precisely determine the timing and temperature constraints of post-impact microbial recolonisation of vug- and fracture-filling assemblages in impactites of the 77.85 ± 0.78 Myr-old Lappajärvi impact crater (Finland).

The earliest evidence of microbial recolonisation was detected through the presence of ^{34}S -depleted pyrite, consistent with sulfate-reducing bacterial (SRB) activity, in minerals precipitated approximately four million years after the impact (73.6 ± 2.2 Ma ago). Clumped isotope analysis showed that the precipitation temperature was about 47 ± 7 degrees Celsius, significantly below the threshold for microbial life. $\delta^{13}\text{C}$ calcite values indicated active methane cycling between methanogens and anaerobic methane-oxidising archaea in symbiosis with SRB during vug mineral formation, which occurred over 10 million years later at progressively lower temperatures.

As concluded by the authors, it would be important to (re)consider previous studies from the Haughton and Chicxulub large impact structures – that suggested ^{34}S -depleted sulfides to be proof of microbial colonization of impact structures within the lifetime of the impact-generated hydrothermal system as these studies did not provide geochronological data and temperature constraints were presented from veins at Haughton only.

The sampling and analytical approaches are sound and I did not find any flaws in the data analysis, interpretation, and conclusions.

REVIEWER COMMENTS

Reviewer #1 (Remarks to the Author):

Overall, this is a well-written, but arguably quite technical, manuscript. Its actual core message, which is to highlight medium-size (and large) terrestrial impact structures as long-lasting hotspots for the emergence and evolution of microbial life in a hydrothermal setting as the crater cools down to ambient conditions, could be conveyed a little more clearly especially in the abstract. The study presents some fascinating (and, as far as I can tell, reliable) U-Pb age results gained from calcite that suggest the crater cooling process lasts, in fact, several million years even in the case of a ~20 km crater – which may be surprising to some (but less so to others). This would make Lappajärvi the single most intensely studied impact crater in terms of geochronology, using different geochronometers and dating minerals, regarding the crater cooling question. It also makes me wonder – if Lappajärvi stays warm for this long – how long will much larger, Chicxulub-size impact craters, take to fully cool down?

If the authors manage to set out their core message and its relevance in a somewhat more striking way, I think publication in Nat. Comm. will be justified. However, looking at the present manuscript I'd rather recommend a more specialized journal, e.g., "Astrobiology", "Earth and Planetary Science Letters", or "Geochimica et Cosmochimica Acta". I also encourage the authors to add a brief discussion of potential mechanisms that could explain the surprising longevity of the heat source or pockets within the crater.

*Response: We thank the reviewer for the insightful comments and the overall positive assessment regarding our study. We agree with the reviewer that our study contributes new conclusions in line with previous studies regarding the longevity the postimpact hydrothermal activity and relatively slow cooling of the Lappajärvi impact crater, and that we go beyond these by tying microbial colonization to the IHG system. Our study thus builds upon and take a significant leap from previous studies rather than contradicts them.

Regarding the general comments on 1) a more striking core message, and 2) the longevity of the heat source, we have made the following amendments, that we believe are in agreement with what the reviewer aimed to achieve by the excellent remarks:

1) In the abstract we changed the following (changes are in bold):

L 24-29: Later stages of vug-mineral precipitation occurred more than **10 Myr** later, at gradually lower temperatures, and featured $\delta^{13}\text{C}_{\text{calcite}}$ values diagnostic for both anaerobic microbial consumption and production of methane. **These new insights confirm a capacity of medium-sized (and large) meteorite impacts in generating long-lasting hydrothermal systems enabling related microbial colonization as the crater cools down to ambient conditions**, that may have important implications for the emergence of life on Earth and beyond.

In the introduction we added a more striking start of the introduction, to even more justify the important context of this work. L32-34: "Meteorite impact craters occur on all planetary bodies in our Solar System. Once thought to be purely destructive events, there is a growing consensus that meteorite impacts may have played an important role in the origin and early evolution of life on Earth^{1,2}. An impact event deposits a tremendous amount of energy in a planetary crust, which results in the fracturing and heating of a huge volume of rock that scales with crater size."

2) The longevity of the heat source is discussed significantly, by including a new paragraph in the text and by adding a long discussion in the Supplementary information, see a response below, to a specific comment.

For the manuscript itself (regardless where it will be published) I recommend minor revision.

Specific comments:

Abstract, line 24: I'd write "10 Myr later", since this describes a time span, and not an age. (I acknowledge some journals only accept Ga/Ma/ka as geologic time units and sometimes change those within the course of the proofing and typesetting process.)

*Response: Thanks, we agree that it should be "10 Myr later", since this describes a time span.

Line 35: Two other suitable references would be Schmieder & Kring (2020, Astrobiology, with a focus on impact ages) and Kenkmann (2021, MAPS).

*Response: Thanks for suggesting these two suitable references. We have added here Kenkmann (2021, MAPS) as it is newer.

L 52: ~1 to 1.6 Myr after... see comment above. "1.6 Ma" is a point in time within the Pleistocene.

*Response: We agree and have changed to "~1 to 1.6 Myr"

L 200/201: Here and in the following lines, the authors report a number of "temperature reconstructions" and choose to label them "temperatures". This sounds a bit like isotope jargon slang and might be misleading to some readers. Perhaps add "...yielded temperature reconstructions (henceforth 'temperatures'), ranging from...". Alternatively, those temperatures could be called paleotemperatures, mineralization temperatures, or simply reconstructed temperatures.

*Response: Thanks for these constructive suggestions, we agree that it might be misleading to only label the temperature reconstructions as "temperatures". In the same paragraph as L200/201, between L217-220, we have changed the following (changed are in bold):

" The clumped isotope analyses yielded **$\Delta 638$ values which were used for temperature reconstructions (hereafter 'temperatures')**, the latter ranging from 8.0 ± 6.8 °C to 47.0 ± 7.1 °C (Supplementary Tables 3, 4). Distinct patterns of **temperatures** are observed between calcite in different $\delta^{13}\text{C}/\delta^{18}\text{O}$ groups (Fig. 5).

L225-229: The $\delta^{18}\text{O}$ composition of the fluids (Supplementary Table 5), from which the calcite precipitated, were calculated based on **clumped isotope temperature reconstructions** and bulk $\delta^{18}\text{O}_{\text{calcite}}$ values by applying temperature-dependent fractionation factors²⁹, showing three distinct $\delta^{18}\text{O}_{\text{fluid}}$ groups with gradually lower temperature (Fig. 5)."

L 326-328: There is something wrong with the sentence "Thermal modelling suggests that cooling below 250 °C took place up to ~1.6 Ma after the impact event¹², although¹¹ estimated slightly faster cooling of >200°C, locally lasting for >1 Myr." Are the authors talking about post-impact cooling rates? Also see comment above regarding Ma vs. Myr.

*Response: Thanks for pointing this out, we agree and have changed to:

L347-350: “Thermal modelling of the post-impact cooling scenario suggests that temperatures below 250 °C were sustained for up to ~1.6 Myr after the impact event¹³, although Kenny et al.¹² estimated slightly faster cooling rates resulting in temperatures >200°C, locally lasting for more than 1 Myr.”

L 333: The authors touch on the geothermal gradient here. This is an interesting aspect. The authors should, at least briefly, discuss the anatomy and role of the central uplift at Lappajärvi, which is not prominent morphologically/topographically as it is concealed by the impact melt sheet. Nevertheless, an impact crater the size of Lappajärvi should be characterized by some ~2.x km of structural uplift within the crater basement at its center. This uplift may also affect the original geothermal stratification within the target rock, i.e., leading to an upward ‘heat bulge’ (compare Naumov, 2005; Schmieder & Jourdan, 2013). Would there be any measurable effect of a ~2 km uplift of the geothermal isoline pattern at Lappajärvi, and could this help explain the ~73 Ma low-temperature ‘warm oasis’ that we see in the calcite U/Pb results? Is Lappajärvi associated with a resolvable geothermal anomaly (compared to the surrounding bedrock) in geophysical maps today?

*Response: These are relevant comments that will be important to address when discussing the prolonged “heat”. We included a discussion about this in the main text:

L369-384: “The longevity of the hydrothermal system at Lappajärvi, or at least pockets of prolonged fluid flow, compared to the similarly-sized Haughton (~50,000 years) and Ries (~250,000 years), may have several explanations. Given the similar size of these three structures, it is unlikely that the elevated geothermal gradient due to impact-induced uplift or residual heat generated by shock and friction within the central uplift would be that different at Lappajärvi. We suggest that the simplest explanation is due to the predominance of crystalline metamorphic rocks in the target rocks at Lappajärvi. This led to the generation of a thick (currently 145 m but originally likely much thicker) layer of silicate impact melt rock, which is the major heat source for hydrothermal systems in mid-size impact structures. In contrast, no coherent bodies of silicate melt rock are known at Haughton and only very small (few m) examples at Ries due to the presence of a thick layer of sedimentary cover rocks at the time of impact. The relatively “dry” crystalline target rocks at Lappajärvi may have also led to relatively slow conductive heat transfer from the crater center, and the low permeability of the target rock could also have prolonged hydrothermal activity (see Supplementary Text 2 for an extended discussion on mechanisms that could explain the longevity of the heat source or pockets within the crater).”

And a significantly longer and more detailed text about this matter in the Supplementary information (too long to include here).

Regarding question 2), we have explored a map of geothermal heat production of the Finnish bedrock (available at Geological Survey of Finland’s map service, <https://gtkdata.gtk.fi/maankamara/>). According to the map, the bedrock at Lappajärvi does not have any anomalous high heat generation. In addition, according to Veikkolainen and Kukkonen (2019), the bedrock itself at Lappajärvi, does not produce anomalous radioactive heat at present. We did not add information about this in the revised manuscript.

L 343: check “a three-stages of evolution”

*Response: Thanks for the comment, we have checked the sentence. “(2005)” was deleted from the reference and we also improved the wording of the sentence.

In addition, Osinski et al.⁵⁵ used fluid inclusions to describe **three-stages of evolution** of the IGH system at Haughton impact structure, which is approximately the same size as Lappajärvi.

Fig. 2d: are the two roundish pyrite grains near the bottom of the right image framboidal pyrite? Or just round crystalline grains? Just as a remark: there are also Fe-Ni-rich metal and sulfide particles in the Lappajärvi melt lithologies that may contain remnants of the impacting projectile (see Fregerslev 1976). Maybe this is worth mentioning.

*Response: Thanks for the interesting remark regarding the pyrite grains, which are actually not framboidal as can be seen in the attached images below. The pyrite fills tiny 10-100 μm -sized, spherical pits at the inner wall of a larger, ~ 0.1 -1mm-sized spherical vesicles within melt glass of the impact melt-bearing breccia. The pyrite grains might appear to be framboidal-like because they fill and are confined by the spherical pits, and could therefore be described to have a spherical shape. In image x, a pyrite grains with probable dodecahedral crystal habit is seen in the upper right corner. In the other image, the pyrite grows in ringlike structures in the pits. We don't have any $\delta^{34}\text{S}$ values of these pyrite, but it would be interesting to measure in the future. Another possibility is that the pyrite could be inclusions of sulfide melt droplets from the impact melt, which makes them out of the scope of the current study, but nevertheless of large scientific interest for future work.

Fig x. pyrite filling tiny 10-100 μm -sized, spherical pits at the inner wall of a larger, ~ 0.1 -1mm-sized spherical vesicles within melt glass of the impact melt-bearing breccia.

Regarding Fe-Ni-rich metal and sulfide mentioned in Fregerslev (1976), we have found a similar spherical grain in the impact melt rock (locally known as “kärnäite”). The composition of this spherule clearly differs from the pyrites of which we measured for sulfur isotope composition of. These sulfide inclusions are a topic of a follow-up study.

We added the following text (ref 30 being Fregerslev, 1976, now added):

L145-148 Zeolites, such as erionite and chabazite (Supplementary Fig. 1a,c), as well as pyrite (Fig. 2d) and Fe-Ni-sulfide aggregates that may contain remnants of the impacting projectile according to previous studies³⁰, are intergrown with impact glass in the impact melt-bearing breccia.

Images of the inclusions with EDS data, for information here only.

Supplementary file and main text: If possible and data exist, the authors could elaborate in a little more detail in which way (if at all) the calcite isotope results are influenced by any internal mineral zonation, e.g., with Fe-, Mn-, and/or Mg-rich zones within the calcite domains (the authors already mention ankerite in the text). At Chicxulub we see an interesting zonation pattern both in SEM-BEI and CL between a Mn-poor and a Mn-rich variety in post-impact hydrothermal sparry carbonate.

*Response: Thanks for suggesting this improvement. Zonation occurs as seen in our BSE-SEM images (Supplementary Fig. 4), of mainly Mn and Fe, and in the CL-images (Mn variability indicated between growth zones). We have added EDS-derived information of the different calcite generations in new figures of the Supplementary information, as well as added some details to the main text regarding compositional differences of the growth zones.

Added figures with compositional data given for different calcite generations:

Calcite i and ii:

Calcite iii:

Calcite y and yy:

A compositional dependence of SIMS $\delta^{18}\text{O}$ and $\delta^{13}\text{C}$ bias might arise from the Fe concentrations, as Fe has been shown to give such an effect, at least for other carbonates (by perhaps 1 permil for $\delta^{18}\text{O}$ at these concentrations, see Rollion-Bard and Marin-Carbonne, 2011), and probably even less for the less studied C-isotope system (not studied for calcite, and hence not discussed in the paper, see e.g. Śliwiński, M.G., Kitajima, K., Kozdon, R., Spicuzza, M.J., Fournelle, J.H., Denny, A. and Valley, J.W. (2016), Secondary Ion Mass Spectrometry Bias on Isotope Ratios in Dolomite–Ankerite, Part II: $\delta^{13}\text{C}$ Matrix Effects. *Geostand Geoanal Res*, 40: 173-184. <https://doi.org/10.1111/j.1751-908X.2015.00380.x> for comparison). We have a secondary control in the form of $\delta^{18}\text{O}/\delta^{13}\text{C}$ from clumped isotope measurements that show overlapping values with the SIMS data (a slight tendency towards higher $\delta^{13}\text{C}$ in the ^{13}C -rich LA1:30 sample is seen in the clumped isotopes, which shows, if anything, that the strong ^{13}C -enrichment is rather understated than overstated). This shows that any compositional dependence on the SIMS data has not affected the values significantly, and certainly not to the level where they affect the process interpretations. For the interpretations of $\delta^{18}\text{O}$ -fluid, we have generally used the $\delta^{18}\text{O}$ from the clumped isotope measurements, so that any minor SIMS-IMF effect is not propagated to those calculations. We added to the method section L543-548: “Calcite trace element concentrations may influence SIMS matrix effects, at least for $\delta^{18}\text{O}^{\text{ref.65}}$, but since 1) the 3-4 mass% of Fe+Mn measured with EDS in the calcite at Lappajärvi (Supplementary Figure 3) may only influence the $\delta^{18}\text{O}$ by a maximum of ~1‰ when applying a correction of Rollion-Bard and Marin-Carbonne⁶⁵, and 2) the clumped isotope measurements and SIMS measurements generally overlap for same samples, we have not applied any correction based on trace element content to the SIMS data.”

Table S6, Excel file: It might be useful to see, in addition to the raw/corrected U/Pb data, individual calculated spot ages with errors, as well.

Apparent age values should be rounded appropriately (I don't think it is useful to report a number as "0.032 +/- 567.610 Ma").

*Response: Although individual ages can be calculated from the ratios, they are meaningless by themselves, and therefore open to possible misinterpretation. We prefer to let readers calculate these if they wish.

The table only includes ratios and not ages. It is standard practice to present the raw isotopic data as they are, even if they seem meaningless. In this case, the quoted number reflects data that is below detection, but that we still tabulate for completeness.

*** End of review ***

Kind regards

Many thanks again for a constructive review

Reviewer #2 (Remarks to the Author):

Congratulations on your excellent contribution to enhancing our understanding of when deep subsurface microbial life was able to recolonize impact-deformed geological formations and their role in deep subsurface elemental cycling (i.e., microbial sulfate reduction and anaerobic methane oxidation).

I have only one comment: Many studies have shown that these microbial processes are ongoing even in nutrient-deprived extreme deep biosphere environments, such as described here. A recent survey from Quraish et al. (2024) in *Geobiology* (not cited) showed that **extant** microbial communities associated with mineral veins and contact zones between breccia and granites, etc., in the granitic basement below the Chicxulub impact crater contain the genetic machinery for chemolithoautotrophic carbon fixation and organoheterotrophy plus cycling of S, Mn, and Fe.

To what extent can the precise dating, identification of minerals, and isotopic analysis on a microscale **exclude the possibility that modern activities caused bias of the isotopic results?** For example, extant chemoautotrophic activities could result in the deposition of microbial carbonates enriched in $\delta^{13}\text{C}$ (overshadowing $\delta^{13}\text{C}$ depleted signals indicative of the onset and extent of methane cycling?). I suggest the authors **discuss this** so that microbiologists interested in this paper know that you have considered it even if it is perhaps a no-brainer for specialist readers that the microscale analysis can rule out that some of the analyzed material was deposited more recently and has no impact on the data or outcomes of the study.

Response:

We thank the reviewer the overall very positive assessment and for the important and insightful comments regarding the potential for extant microbial biosignatures (chemolithoautotrophic carbon fixation and organoheterotrophy plus cycling of S, Mn, and Fe) to be associated with mineral veins and contact zones between impact breccia and target rocks.

We have reasons why we are confident that modern activities have not caused any bias of our isotopic results. First, we made sure to measure the isotopic ratios with SIMS well within the mineral crystals. Secondly, we have robust U-Pb dating on some of the calcite crystals, which exclude that possibility that modern activities caused bias of the isotopic results...In some calcite crystals AOM-signatures were measured in the outermost part of calcite crystals. Since no robust U-Pb dating were made on the outermost part or the latter crystals, modern activities cannot be ruled out as being responsible for the AOM-signatures, although not highly likely even in that case, as that growth zone show evidence of precipitation (sharp border to earlier growth phase) from solution and not dissolution-reprecipitation that is more likely if there is a modern influence on any of the mineral coatings. We have added some text on this matter to highlight that we use mineral interiors to detect ancient biosignatures (introduction: “Using micro-scale isotopic techniques **of mineral interiors**, we provide direct evidence of **ancient** microbial colonization during the waning stages of the IGH system of this terrestrial impact structure”, and we have added the reference, as we believe that it is an important study to mention in the introduction, and we thank the reviewer for notifying us about this study. We added this: “At the Chicxulub impact crater, microscopic pyrite framboids with $\delta^{34}\text{S}$ values as low as -35‰ and $\Delta\text{S}_{\text{sulfate-pyrite}}$ values up to 54‰ , are also consistent with MSR²⁴, and studies of extant microbiological communities show that the impact-induced geochemical boundaries have shaped the modern-day deep biosphere in the granitic basement underlying this crater²⁵.” Reference 25 being Quraish et al. (2024) in *Geobiology*, now cited.

Reviewer #3 (Remarks to the Author):

This is a very interesting paper that I thoroughly enjoyed reading. I appreciate the coordination between stable isotope and radiometric analyses to understand the timing of microbial colonization as it is linked (or not linked) to impact-generated hydrothermal systems. This specific type of work in the field of terrestrial impact cratering, and particularly at the intersection of astrobiology, is sorely needed and is highly relevant as we continue to explore craters on Earth and the ancient Martian surface (and potentially subsurface, in the future!)

The work presented here addresses a key question related to impact-generated hydrothermal (IGH) systems which is **timing**, as it relates to post-impact colonization by microbial communities. Previous studies on sulfide mineral isotopes, for example, in impact craters have identified biosignatures in the form of highly reduced (isotopically light) $\delta^{34}\text{S}$, implying a biogenic rather than (abiotic) thermochemical sulfate reduction origin, but the timing of this activity has remained ambiguous and often linked directly to the hydrothermal system without more definitive (e.g., radiogenic isotope) constraints.

**Response: We acknowledge the reviewer for an overall positive assessment and for highlighting the greater implications of our work. The detailed comments below are all addressed, and have improved the revised version of the manuscript significantly.*

I have a few comments on general observations of the manuscript as well as line-by-line suggestions, below.

Sections of the methods and results related to mineral identification and lithologic context – and by extension the corresponding isotopic analyses - are unclear and would benefit from better organization and clarification. I find it difficult to interpret the isotope data without a better understanding of the context and what exactly has been analyzed. Are many of these generational relationships implied based solely on BSE images of rough sample surfaces? Is there supporting chemical data (e.g., something that clearly shows different generations of calcite) or possibly thin section images to support these implied relationships? If that data is available it would be immensely helpful for the reader and solidify some of these interpretations. I understand this paper represents a lot of coordination between in-situ and bulk analyses, and care should be taken to make it clear what has been analyzed, actually identified and what it represents.

**Response: Indeed, the paper summarizes a plethora of different techniques and materials, and we have been very careful to control how these are interconnected. We agree that some of this information could be better emphasized and more thoroughly presented in the manuscript, and we have made a comprehensive effort include these in the revised manuscript. We acknowledge the reviewer for these important constructive remarks, that we believe have significantly improved the scientific presentation, and we believe that we have now reached a level of clarity that the reviewer aimed to achieve. For a detailed representation see responses to comments below. For instance, calcite generations are interpreted based on CL-images and BSE-images of polished crystal interiors of minerals on fracture surfaces and indeed also in thin sections and polished rock blocks analyzed by SEM (with EDS compositions for different minerals and generations, as well as EDS-maps for key features, now added). These crystals are of various homogeneity and were targeted by micro analysis for stable isotopes (SIMS) and radiogenic isotopes (LA-ICP-MS) and analysed in bulk or in subgenerations (micro-drilled) for clumped isotopes. The relationships of these analyses are now better defined, see below.*

Are many of these generational relationships implied based solely on BSE images of rough sample surfaces- For example, to examine the identification of a “serpentine-group clay” (Lines 122, 126, 144) I would expect XRD data: I see one XRD pattern - supplemental Fig. 9 - with no labeled peaks or mineral ID’s, and S8_XRD representing only one analysis, with a list of potential library matches, but I see no labeled clay (or other mineral) d-spacings indicative of serpentine, vermiculite or corrensite 001’s, 060’s. Nor do the methods describe the careful analyses to identify corrensite, serpentine or vermiculite, a high charge smectite, were undertaken. The caption also states this was the clay coating on the vug, in the hand sample, but it’s unclear how this context was determined - if this is a bulk, crushed sample of a melt rock it could have been a secondary clay in the matrix, or something else as clays can form from altering many of the primary aluminosilicate phases under hydrothermal conditions.

Response: We have analyzed thin sections, polished rock blocks and uncoated open vugs under petrographic and stereomicroscope as well as SEM-EDS, and hand-picked minerals, that were polished to reveal cross-sections, and these cross-sections (calcite and pyrite) were also analyzed in detail with CL and SEM-EDS to reveal growth zones, that were then targeted with SIMS spot analysis and LA-ICP-MS. We used a combination of petrographic observations, SEM-EDS, CL and SIMS data to define the generations of calcite presented. We have added a series of highly detailed images that combines petrographic and EDS data with sketches of how minerals were sampled and later prepared and analyzed in microscale (and in bulk of subsamples) in Suppl. Figures 3-6 (3-5 enclosed below as well). For the XRD, we have toned down some of these interpretations, because we cannot distinguish between corrensite and smectite with conventional methods as the reviewer notes. These observations are not critical for the main scope of this paper, but will be described more in depth in a future study focused on clay mineralogy, lead by G. Osinski’s team. The XRD-analyses of the current manuscript are done on mineral separates, scraped of the fracture- and vug surfaces, and hand-drilled sample from a vein. We have further added an XRD-spectrum of siderite and calcite. We have added a description of how the minerals were sampled for XRD to the method section in the Suppl.information. For vermiculite and for the siderite/calcite XRD data, we have updated the peaks in the supplementary information (S8_XRD) so that it is clear how peaks are labeled and assigned, and references to accompany them. We agree that vermiculite can only be identified with certainty using a more detailed clay mineral-specific XRD protocol, that can determine swelling components etc, and therefore (in addition to that the vermiculite seems to be of low crystallinity), we have termed it vermiculite throughout the manuscript and described the uncertainty of this identification.

Added figures with compositional data given for different calcite generations:

Calcite i and ii:

Calcite iii:

Calcite y and yy:

- As another example, I see lots of beautiful photographs and BSE images of rough sample surfaces but no thin sections or chemical (EDS) data to support these interpretations. If this is available (Lines 461-464 suggest EDS analyses should exist), please add a few figures (e.g., chemical maps, spot analyses, something to help place the isotopic analyses and mineral, lithologic relationships into context).

*Response: As described in the comment above, we do not base our interpretations solely on photographs and BSE images of rough sample surfaces, but also on BSE images and EDS data of polished rock surfaces (showing trace element differences between calcite generations for instance), polished thin sections, and EDS and SIMS data of polished pyrite- and calcite crystals in epoxy mounts. For SIMS, we need to use epoxy mounts with a large number of individual calcite and pyrite crystals that are analyzed in an automated routine. Hence, we were able to gather a substantial number of $\delta^{13}\text{C}$, $\delta^{18}\text{O}$ and $\delta^{34}\text{S}$ values, which would not have been possible if using thin sections (in fact thin sections cannot be used easily in SIMS, and although it may have made the petrographic provenance more robust, it would have decreased the number of spots by 90%, so it was an easy choice and we made sure to have strong petrographic documentation of the mounted, hand-chiseled grains). We have added more information (Suppl. Information Fig 3-6) and added EDS spectra and EDS-maps for key petrographic relation features.

- As a final example, the context for many of these minerals used for isotopic analysis is unclear. E.g., It appears there are more than one occurrences of pyrite in these samples (Fig. 2a vs 2d). Where are the pyrite crystals shown in Supplemental Fig. 3 (and by extension, the longer list of sample ID's in the excel sheet) from, other than "the melt rock" or "the lithic impact breccia?" Are these fracture coatings or vug-fillings (e.g., Fig. 2)? Are they potentially hand-chiseled out (Lines 466-468) from the melt or breccia matrix – some of these are 100 microns or less, as low as 10 microns (Line 134) in size? These relationships really matter when it comes to interpreting their isotopes, particularly as there is such an unusually large range of data here. I have no doubt that with the amount of work

performed here, the data confirming these mineral identifications and lithologic, contextual relationships is available to show the reader?

*Response: We have replied to most of this to the comments above. But, in addition, we have added more context. Like adding information to caption of Supplemental Figure 3, “from pyrite intergrown with the first generation of calcite in this sample (i.e. with calcite group i in Fig. 2a of the main text). For 3a”. And “petrographic context shown in Supplementary Fig 2b” for 3b. We have also added the detailed figures on how samples are analyzed, from drill core to microscale, and added EDS spectra to them as well. The samples are from both veins and vugs, as shown in the images. We have added host rock/type of sample to the Suppl lists of analyzes.

This is exciting work and I hope the authors can take the steps to organize, present, coordinate and clarify their methods and results, especially the implied mineral identifications, relationships and context. This is a really interesting study and I would look forward to seeing some of this information clarified to better piece together the story presented here!

*Response: Once again, thanks again for the overall positive assessment and constructive comments that will improve the paper significantly. We have made a substantial effort to comply with these comments in our revised manuscript.

Line by line comments:

Lines 49-52: As written it is unclear whether this statement implies that hydrothermal circulation and cooling below 250 degrees began at 1-1.6 Ma or lasted this long. Please clarify.

*Response: corrected, as also replied to reviewer 1.

Line 53: Minor typo: “...modeling have predicted...” to “...modeling has predicted...”

*Response: Changed from “...modeling have predicted...” to “...thermal models have predicted...”

Lines 75-78: I would argue here that Chicxulub is an exception - those frambooids are intergrown with the dachiardite and analcime - these zeolites are high(er) temperature hydrothermal minerals, supporting these LIKELY formed during a period of elevated temperatures. The analcime samples in particular - the images in Kring et al. (2021), and the mineralogical, in-situ context of those crystals - it's hard to make the argument that the frambooids formed much later, during diagenesis/far below and outside IGH conditions.

*Response: Although we don't argue with that interpretation, the point we make here is that no geochronological dating presented for the Chicxulub mineral assemblages. The point made in the introduction is that in situ dating is required, especially since we show ourselves that mineral filled-vugs that are completely sealed and residing in impact glass may have ages being far younger than the impact. Without any geochronology evidence, we would probably have interpreted them as primary and impact related. Hence, we did not change the text, the reader can go back to the original paper and make their own assessment on the Chicxulub mineral assemblages.

Lines 80-85: Very minor comment – I agree with the overall statement – that there is a need for timing constraints with the biosignature examples described in the previous section, but the biosignatures described in this specific section are very different from the ones listed in the previous paragraph, which is a tad misleading (e.g., 34S of sulfides attributed to bacterial sulfate reduction at elevated

temperatures). I would also suggest adding a couple words here that indicate these biosignatures were initially attributed directly to impact hydrothermal activity. When I first read this I thought “well yeah these old rocks are probably covered in some sort of fungus” but went to the reference (and was surprised) to see someone initially claimed these were hydrothermal system-hosted fungi! Similar comment for calcite, which can easily be reset and even initially form at surface diagenetic conditions.

*Response: Yes, agreed it needs more context, which we added (bold), L88-91: “Using *in situ* Rb/Sr dating of calcite-albite-K feldspar assemblages occurring alongside fossilized fungi, that were initially attributed to colonization during impact-related hydrothermal activity²⁶, Tillberg et al.²⁷ found that fungal colonization of the 458 Ma Lockne impact structure occurred at least 100 million years after the impact” Ref 26 being added Ivarsson et al.

Lines 97-98: The comment on methane seepage is a bit out of place. I understand it’s relevance but I would suggest removing this entirely or elaborating on how this is tied to the rest of the sentence and section (e.g., are the authors implying the methane seepage was directly linked to IGH system-hosted microbes?) I would err on the side of caution here, as the source of Martian methane detections are a highly contested topic, and very unlikely to be IGH system related.

*Response: Agreed, and if we were to elaborate on this, it would perhaps be too extraneous. We removed that part of the text L106: “~~methane seepage has been detected in the Gale crater on Mars~~”²⁶.”

Line 131-133: “Clay minerals, zeolite and pyrite were among the first mineral coatings to appear intergrown with impact glass (Supplementary Fig. 1a,c)” Clay minerals and pyrite are not the first secondary minerals to form in these systems – they are often later stage, especially clays. What is the evidence for this, based only on the BSE images? I also see no clays labeled or identified here. Is there EDS data to go with these images?

Response: Agreed with respect to clay minerals. There is a clay mineral, plausibly vermiculite in Fig. 2b, but we don’t know how old this mineral is but it is likely not one the first secondary minerals to form in these systems. “Clay” is erased from the sentence. The sentence has been changed to L145-148: “Zeolites, such as erionite and chabazite (Supplementary Fig. 1a,c), as well as pyrite (Fig. 2d) and Fe-Ni-sulfide aggregates that may contain remnants of the impacting projectile according to previous studies³⁰, are intergrown with impact glass in the impact melt-bearing breccia.” We also have added EDS data to the supplementary information, as detailed above. And we have given details on the “vermiculite” naming, previously and at other place in the manuscript.

Line 132: Which zeolite? Or change to zeolites.

*Response: Change to “zeolites.”

Line 140, Figure 2: Is there a difference in composition between these different types of calcite?

*Response: We have given some more information on this in the text and added a lot of information in the supplementary information. Calcite I and ii are more Fe rich (albeit low %) than y and yy for instance.

Line 143: Where is the data indicating this is ankerite? I see no labeled XRD patterns or in-situ EDS data on thin sections?

*Response: We have done a complete overhaul of this following these detailed comments (thanks for asking!). We have done substantially more SEM-EDS-work and found out that ankerite is very minor and siderite is much more abundant. Hence, we are not presenting ankerite as a certain identification any longer, but have instead added siderite (confirmed by EDS spectrum and XRD spectrum). We have also added EDS data for the different calcites, and pyrite.

Lines 149-150: What pyrite is being analyzing here exactly?

*Response: This text was added: L162-163: “of all pyrite crystals analyzed by SIMS in this study”

Lines 163-170, Figure 3: what is the context for these pyrite grains (e.g., matrix, fracture or vug coating?) Additionally, what is the other mineral in 3b?

*Response: 3a: fracture/vein, 3b: matrix, 3c: vug. We have added this information to the caption.

The other mineral is calcedony. We have EDS spectra for it, below. And have added this information to the caption. “showing pyrite along with calcedony (BSE-dark mineral)”

Lines 192-193: How are these petrographically identified as chronologically older? Based on BSE images from rough sample surfaces? Are there thin sections (and corresponding EDS analyses) of these samples?

*Response: These groups are visible in CL, BSE and/or indicated by SIMS data. From thin section and mainly from polished cross sections of individual grains targeted by SIMS spots. We added this information to the caption.

Lines 271-275: These are very unusual and heavy values for sulfides – a very interesting and exciting find! I see examples of these isotopically heavy sulfides in Fig. 3b and c – any chance you also have corresponding EDS data on these? Or the other mineral intergrown with the pyrite in 3b. I’m curious to know if there’s a chemical difference or zoning – anything – that might help interpret these unexpected results.

*Response: We do not see any compositional difference distinguishable with EDS. All spectra show just Fe and S. However, high resolution methods like LA-ICP-MS might be able to distinguish such variations. We aim to explore this in a future study.

Lines 452-464: I see methods listed here for low vacuum analyses on rough sample surfaces. Is this what the chronological relationships and mineral identification, context is based on? Other than a single sample bulk XRD analysis and the second set of “SEM analysis” of hand-chiseled and epoxy

mounted minerals listed 466-471? Once the samples are hand chiseled you lose a lot of contextual information. I'm curious why no thin section petrographic work was completed. If it was (e.g., EDS mapping at high vacuum, or spot analyses and imaging to understand the in-situ mineralogical, chemical context), please add it to the manuscript.

*Response: As stated above we did use thin sections as well, and have added this information to the method section, and to the Supplementary information (including EDS mapping).

Supplemental Fig. 1,2: I see several BSE images showing beautiful mineral surfaces, and vugs but no chemical (e.g., EDS, WDS) or crystallographic data (e.g. XRD, outside the unlabeled Supp. Fig. 9). Supplemental Fig. 9: Please label mineral peaks here (e.g., mineral ID and their respective, indicative 001, 060 d-spacings). How was vermiculite characterized? Clay minerals are notoriously difficult to identify – vermiculite is a high-charge smectite. Is there additional information that shows this is indeed vermiculite? What about the other clays listed in the manuscript – serpentine and corrensite?

*Response: This is a very relevant comment, and we have added more information, as outlined in comments above, and added peaks to the XRD spectrum, and added another XRD spectrum, as well as many EDS spectra and maps. We have also toned down the clay mineral identifications, to being more “indications” than identifications.

Remaining reviewer comments

Our guidance:

Your response:

R1 : having reviewed the second round of this manuscript submission I think the paper is now good to go, with a few final suggestions listed below.

L 29. I'd write „..., which may have...”

L66/L67: 2 x “have been reported”

L72: ...are also...

L97: Yes, but there may be other contexts, e.g., hydrothermal mineral assemblages and sequences with traces of colonization linked to specific minerals using purely petrographic methods, providing a relative timeline of colonization.

L121: maybe remove “unshocked” and replace by “fractured” – this unit was presumably also shocked, but at low levels not immediately recognizable in terms of shock metamorphism.

General: Please proofread a final time for the filling word ‘also’ – there is (locally) an overabundance. Perhaps replace by “Moreover,” or “as well” where appropriate.

L195: 4 cm-long or 4 cm-diameter vug?

L251: comma after parentheses

L300: are consumed?

L491: Sentences should not start with a number. Perhaps write out “Thirty-three...”

Kind regards,
Martin Schmieder

We thank the reviewer for the positive assessment of the second round of this manuscript submission. We also thank the reviewer for the final suggestions, which we have corrected:

L 29. We added “an effect” before “that” in the sentence, so that it is grammatically correct:

“These insights confirm the capacity of medium-sized (and large) meteorite impacts to generate long-lasting hydrothermal systems, enabling microbial colonization as the crater cools to ambient conditions, an effect that may have important implications for the emergence of life on Earth and beyond.”

L66/L67: “has” is grammatically correct, so we did not change anything here.

L72: We added “are” and improved to sentence to: “Likewise, at Rochechouart, $\delta^{34}\text{S}$ sulfide values down to -36‰ are consistent with MSR²²”

L 97: We changed to: “Radioisotopic dating, along with other contexts, such as petrographic relations, is essential to verify such colonization.”

L121/L472: We agree, “unshocked” has been replaced by “fractured”

General comments about the filling word “also”:
We have replaced “also” by “moreover” or “as well” etc. where appropriate.

L195: We changed to “4 cm diameter vug”

L251: We added a comma after the parentheses

L300: “is” is grammatically correct

L491: We agree and changed to “Thirty-three...”